# ROUTING MATTERS IN MOE: SCALING DIFFUSION TRANSFORMERS WITH EXPLICIT ROUTING GUIDANCE

**Yujie Wei**[1], **Shiwei Zhang**[2*], **Hangjie Yuan**[3], **Yujin Han**[4], **Zhekai Chen**[4,5], **Jiayu Wang**[2],
**Difan Zou**[4], **Xihui Liu**[4,5], **Yingya Zhang**[2], **Yu Liu**[2], **Hongming Shan**[1†]

[1]Fudan University   [2]Tongyi Lab, Alibaba Group   [3]Zhejiang University
[4]The University of Hong Kong   [5]MMLab
yjwei22@m.fudan.edu.cn, zhangjin.zsw@alibaba-inc.com, hmshan@fudan.edu.cn

## ABSTRACT

Mixture-of-Experts (MoE) has emerged as a powerful paradigm for scaling model capacity while preserving computational efficiency. Despite its notable success in large language models (LLMs), existing attempts to apply MoE to Diffusion Transformers (DiTs) have yielded limited gains. We attribute this gap to fundamental differences between language and visual tokens. Language tokens are semantically dense with pronounced inter-token variation, while visual tokens exhibit spatial redundancy and functional heterogeneity, hindering expert specialization in vision MoE. To this end, we present **ProMoE**, an MoE framework featuring a two-step router with explicit routing guidance that promotes expert specialization. Specifically, this guidance encourages the router to *first* partition image tokens into conditional and unconditional sets via conditional routing according to their functional roles, and *second* refine the assignments of conditional image tokens through prototypical routing with learnable prototypes based on semantic content. Moreover, the similarity-based expert allocation in latent space enabled by prototypical routing offers a natural mechanism for incorporating explicit semantic guidance, and we validate that such guidance is crucial for vision MoE. Building on this, we propose a routing contrastive loss that explicitly enhances the prototypical routing process, promoting intra-expert coherence and inter-expert diversity. Extensive experiments on ImageNet benchmark demonstrate that ProMoE surpasses state-of-the-art methods under both Rectified Flow and DDPM training objectives. Code is available at https://github.com/ali-vilab/ProMoE.

## 1 INTRODUCTION

Diffusion models (Ho et al., 2020) have made substantial advances for visual synthesis (Rombach et al., 2022b; Yang et al., 2024; Wei et al., 2024a; Wan et al., 2025). Driven by the growing demand for higher fidelity and quality, research has focused on scaling up diffusion models (Esser et al., 2024b) and propelled an architectural shift from U-Net (Ronneberger et al., 2015) backbones to the now-prevalent Diffusion Transformers (DiTs) (Peebles & Xie, 2023). Despite the proven effectiveness of DiT-based models (Esser et al., 2024a), their dense activation of all parameters irrespective of task or input incurs substantial computational overhead, thereby hindering further scalability.

To scale toward larger and more capable models, the large language model (LLM) community has widely adopted the Mixture-of-Experts (MoE) (Jacobs et al., 1991; Shazeer et al., 2017) paradigm, which expands model capacity while maintaining computational efficiency. Conceptually, an MoE layer dispatches input tokens to specialized "expert" sub-networks via a router and

---

*Project Leader   † Corresponding Author

Yujie Wei and Hongming Shan are with Institute of Science and Technology for Brain-inspired Intelligence, MOE Frontiers Center for Brain Science, Key Laboratory of Computational Neuroscience and Brain-Inspired Intelligence, and State Key Laboratory of Brain Function and Disorders, Fudan University, Shanghai, China

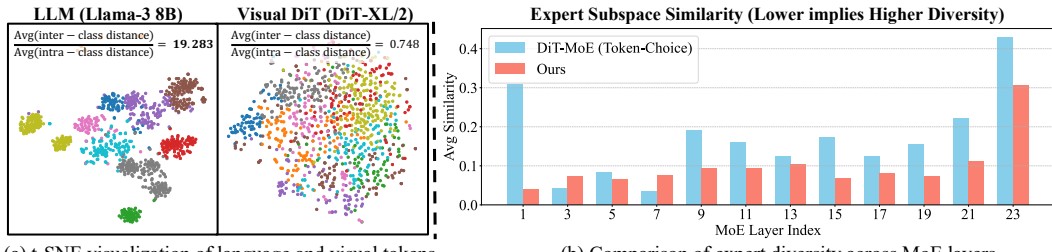

Figure 1: (a) We randomly sample 1k intermediate-layer tokens from 110 ImageNet classes for 10-cluster k-means clustering (differentiated by color). With class names/labels as inputs, LLM tokens form compact, well-separated clusters with high semantic density, whereas visual tokens are diffuse. This disparity is quantified by the ratio of inter- to intra-class distance ($19.283 \gg 0.748$). (b) We measure inter-expert diversity using singular value decomposition on each MoE layer's expert weight matrices and computing the mean similarity of the subspaces spanned by their top-k left singular vectors (Hu et al., 2021). Incorporating routing guidance (Ours) enhances expert diversity.

returns a weighted sum of the selected experts' outputs. Despite MoE's profound success in language modeling (Jiang et al., 2024; Dai et al., 2024), recent efforts to integrate it into DiT models have not yielded the significant gains observed in LLMs. Specifically, DiT-MoE (Fei et al., 2024), which employs token-to-expert routing, often underperforms dense counterparts despite activating the same number of parameters. In contrast, EC-DiT (Sun et al., 2024), which assigns each expert a fixed quota of tokens, delivers only marginal gains even with extended training. More recently, DiffMoE (Shi et al., 2025), which introduces a global token-distribution routing scheme, still reports relatively limited improvements. This pronounced gap between MoE's transformative impact in LLMs and its modest returns in DiT models motivates a fundamental question: *What are the underlying factors that impede the effectiveness of MoE in DiT models?*

To answer this question, we examine how linguistic and visual inputs differ in models and highlight the following two distinctive properties of visual inputs. **1) High Spatial Redundancy.** Unlike discrete text tokens, which are semantically dense with salient inter-token differences, visual tokens (*i.e.*, image patches) are continuous, spatially coupled, and substantially redundant (Fig. 1(a)). The high correlation between patches often leads experts to learn homogeneous features. **2) Functional Heterogeneity.** The practice of classifier-free guidance (Ho & Salimans, 2022) in diffusion models inherently introduces two functionally distinct input types: conditional and unconditional. A naive MoE treats them uniformly with undifferentiated routing, ignoring their different roles. These properties collectively impede effective expert diversity and specialization (Fig. 1(b)).

Motivated by these observations, we revisit the foundational principle of MoE design: expert specialization, in which each expert acquires focused and non-overlapping knowledge (Dai et al., 2024; Cai et al., 2025). We decompose this objective into two criteria: **Intra-Expert Coherence**, which ensures that an expert consistently processes similar patterns, maintaining a stable functional role; and **Inter-Expert Diversity**, which encourages different experts to specialize in distinct tasks to achieve functional differentiation. In language modeling, the semantic density and separability of discrete text tokens provide a potent inductive bias that naturally fosters expert specialization, satisfying both criteria. In contrast, for visual inputs, the combination of intrinsic redundancy and extrinsic functional heterogeneity makes expert specialization non-trivial. Therefore, in this paper, we move beyond implicit expert allocation, and *introduce explicit routing guidance designs to promote both intra-expert coherence and inter-expert diversity*.

To this end, we present **ProMoE**, a Mixture-of-Experts framework featuring a two-step router with explicit routing guidance to promote expert specialization. Specifically, this guidance provides two distinct routing signals: the token's functional role and its semantic content. Guided by these signals, the router implements two steps: *conditional routing* and *prototypical routing*. *First*, conditional routing addresses functional heterogeneity by partitioning visual tokens into unconditional and conditional sets. Unconditional image tokens, derived from image patches under null conditioning (*e.g.*, empty labels or texts), are processed by dedicated *unconditional experts*. In contrast, conditional image tokens, obtained from patches under specific conditioning, are dispatched to standard experts (routed experts). This hard routing mechanism enforces functional segregation, fostering specialization across unconditional and routed experts. *Second*, prototypical routing further assigns conditional image tokens using a set of learnable prototypes, each associated with a specific expert, by computing cosine similarity between token embeddings and the prototypes in latent space.

While prototypical routing is flexible and effective, it still relies on implicit learning from token semantics. Fortunately, its similarity-based allocation in latent space provides a natural mechanism for injecting explicit semantic routing guidance. We validate the importance of semantic guidance in systematic experiments (Sec. 4.2), where both explicit (classification-based) and implicit (clustering-based) guidance yield clear improvements. Building on this, we propose a *routing contrastive loss* that explicitly enhances the prototypical routing process by assigning semantically similar tokens to the same expert while preserving distinct token distributions across experts. Compared with alternative guidance strategies, the proposed contrastive loss requires no manual labels and is more robust, promoting intra-expert coherence and inter-expert diversity in vision MoE.

Extensive experimental results demonstrate ProMoE's superior performance and effective scalability on both Flow Matching and DDPM paradigms. Notably, ProMoE achieves significant gains over dense models despite using fewer activated parameters, and surpasses state-of-the-art methods that have $1.7\times$ more total parameters than ours.

In summary, our contributions are fourfold: *1)* By analyzing differences between language and visual tokens, we present **ProMoE**, an MoE framework with explicit routing guidance for DiT models. *2)* We design a two-step router, where conditional routing first partitions image tokens by functional roles, and prototypical routing then refines assignments using learnable prototypes based on semantic content. *3)* We propose a routing contrastive loss that enhances prototypical routing, explicitly enforcing intra-expert coherence and inter-expert diversity. *4)* Extensive experiments demonstrate that ProMoE outperforms dense models and state-of-the-art MoE methods across diverse settings.

## 2 RELATED WORK

**Diffusion Models.** Diffusion models (Nichol & Dhariwal, 2021; Wu et al., 2025b; Deng et al., 2026) have made remarkable progress in visual synthesis. Early work (Rombach et al., 2022a; Podell et al., 2023; Wei et al., 2024b) primarily use U-Net trained with the DDPM objective (Ho et al., 2020; Song et al., 2020). Recent models (Chen et al., 2023; Ma et al., 2024; Hatamizadeh et al., 2024; Chu et al., 2024; Wu et al., 2025a; Wei et al., 2025; Tang et al., 2025) have shifted to Transformer architecture, offering superior scalability and generative quality, and many are trained with the more effective Rectified Flow (RF) (Liu et al., 2022), a flow-matching formulation (Lipman et al., 2022) that constructs a straight-line path between data and noise distributions. In this work, we adopt a standard DiT backbone and train with both DDPM and RF objectives, demonstrating the effectiveness and scalability of our approach across different training paradigms.

**Mixture of Experts.** Mixture-of-Experts (MoE) (Jacobs et al., 1991; Shazeer et al., 2017; Lepikhin et al., 2020) are designed to expand model capacity while minimizing computational overhead by sparsely activating sub-networks for distinct inputs. Inspired by MoE successes in LLMs (Dai et al., 2024; Liu et al., 2024; Li et al., 2025; Muennighoff et al., 2024), recent work has integrated MoE to scale diffusion models to improve generative quality (Riquelme et al., 2021). Early MoE applications in U-Net-based diffusion models (Lee et al., 2024; Balaji et al., 2022; Feng et al., 2023; Xue et al., 2023; Park et al., 2023; 2024; Zhao et al., 2024) often assign experts by diffusion timestep ranges, showing strong scaling potential. However, adapting MoE to DiT architecture (Shen et al., 2025; Sehwag et al., 2025; Cheng et al., 2025) faces several limitations. Token-choice routing methods (*e.g.*, DiT-MoE (Fei et al., 2024)) suffer poor expert specialization due to imbalanced token assignments, whereas expert-choice methods (*e.g.*, EC-DiT (Sun et al., 2024)) that fix token quotas per expert yield only marginal gains. More recently, DiffMoE (Shi et al., 2025) and Expert Race (Yuan et al., 2025) explore batch-level global token selection and mutual expert–token routing, yet still rely on implicit expert learning and struggle with limited expert specialization due to the redundancy and functional heterogeneity of visual tokens. In contrast, we analyze language–vision token differences and introduce explicit routing guidance to the MoE router based on the token's functional role and its semantic content. We further enhance the routing process through the proposed routing contrastive loss, promoting intra-expert coherence and inter-expert diversity.

## 3 PRELIMINARIES

**Diffusion Models.** Diffusion models are generative models that learn data distributions by reversing a forward noising process. The continuous-time forward process can be formulated as

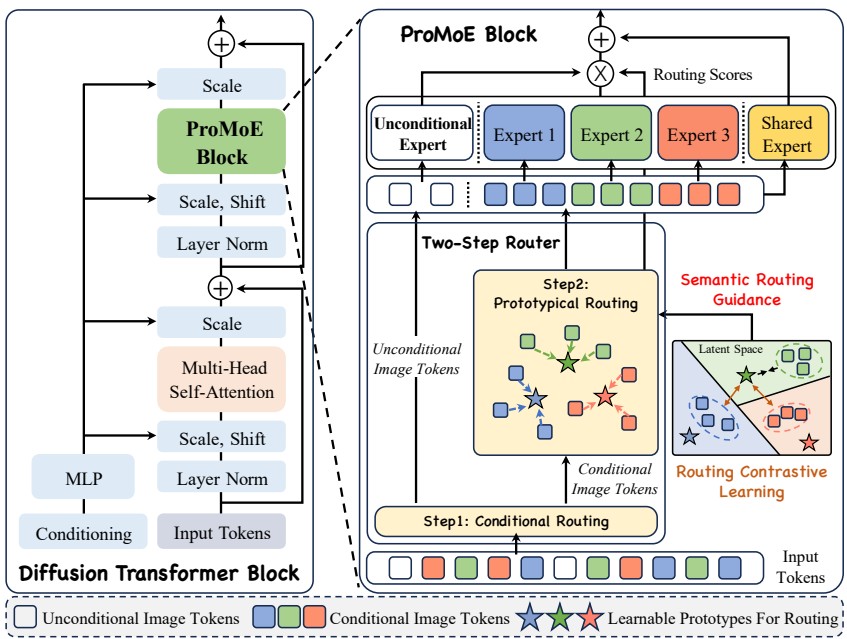

Figure 2: **Overview of ProMoE architecture.** The input tokens are split by conditional routing into unconditional and conditional subsets. Unconditional image tokens are processed by unconditional experts. Conditional image tokens are assigned by prototypical routing with learnable prototypes. The routing contrastive learning explicitly enhances semantic guidance in prototypical routing.

$\mathbf{x}_t = \alpha_t \mathbf{x}_0 + \sigma_t \boldsymbol{\epsilon}$, with $t \in \mathcal{U}(0,1)$ and $\boldsymbol{\epsilon} \sim \mathcal{N}(0, \mathbf{I})$. $\alpha_t$ and $\sigma_t$ are monotonically decreasing and increasing functions of $t$, respectively. For the reverse process, a denoising network $\mathcal{F}_\theta$ is trained to predict the target $\mathbf{y}$ at each timestep $t$, conditioned on $\mathbf{c}$ (*e.g.*, class labels or text prompts):

$$\mathcal{L} = \mathbb{E}_{\mathbf{x}_0, \mathbf{c}, \boldsymbol{\epsilon}, t} \left[ \|\mathbf{y} - \mathcal{F}_\theta (\mathbf{x}_t, \mathbf{c}, t)\|_2^2 \right], \qquad (1)$$

where the training target $\mathbf{y}$ can be the Gaussian noise $\boldsymbol{\epsilon}$ for DDPM models (Ho et al., 2020), or the vector field $(\boldsymbol{\epsilon} - \mathbf{x}_0)$ for Rectified Flow models (Liu et al., 2022).

**Mixture of Experts.**   Mixture-of-Experts (MoE) is an architectural paradigm that scales model capacity while preserving computational efficiency by selectively activating a subset of "experts" sub-networks. A standard MoE layer comprises $N_E$ experts and a trainable router $\mathcal{R}$. Each expert $E_i$ is implemented as a feed-forward network (FFN). Given input $\mathbf{X} \in \mathbb{R}^{B \times L \times D}$, where $B$ is the batch size, $L$ is the token length, $D$ is the hidden dimension, the router $\mathcal{R}$ maps the input $\mathbf{X}$ to token–expert affinity scores $\mathbf{S} \in \mathbb{R}^{B \times L \times N_E}$ via an activation function $\mathcal{A}$:

$$\mathbf{S} = \mathcal{A}(\mathcal{R}(\mathbf{X})). \qquad (2)$$

At each forward pass, the router activates the top-$K$ highest-scoring experts and dispatches the input to them. The final output is the weighted sum of the activated experts' outputs with a gating function:

$$\mathbf{G} = \begin{cases} \mathbf{S}, & \text{if } \mathbf{S} \in \text{TopK}(\mathbf{S}) \\ 0, & \text{Otherwise} \end{cases}, \quad \text{MoE}(\mathbf{X}) = \sum_{i=1}^{N_E} \mathbf{G}_i * E_i(\mathbf{X}), \qquad (3)$$

where $\mathbf{G} \in \mathbb{R}^{B \times L \times N_E}$ is the final gating tensor. There are two common routing paradigms in MoE: Token-Choice (TC) and Expert-Choice (EC). In TC, each token independently selects its top-$K$ experts; in EC, each expert selects a fixed number of top-$K$ tokens.

## 4   PROMOE

In this section, we present ProMoE, an MoE framework for DiTs that integrates a two-step router with explicit routing guidance. The overall pipeline is depicted in Fig. 2. We first detail the two-step router in Sec. 4.1. We then validate the importance of semantic routing guidance in visual MoEs in Sec. 4.2 and further propose routing contrastive learning to enhance semantic guidance in Sec. 4.3.

## 4.1 Two-Step Router

The ProMoE router operates in two steps: conditional routing based on the token's functional role, followed by fine-grained prototypical routing based on token semantics.

**Conditional Routing.** Unlike LLMs, diffusion models typically employ classifier-free guidance (CFG) (Ho & Salimans, 2022) at inference to enhance sample quality. Specifically, CFG steers the generation process by combining the model's conditional and unconditional noise predictions. This paradigm naturally defines two functionally heterogeneous tokens: 1) unconditional image tokens, derived from image patches under null conditioning (*e.g.*, empty labels or texts); and 2) conditional image tokens, obtained from patches under specific conditioning (*e.g.*, class labels or texts).

To handle different token types, the first step of the ProMoE router employs hard routing based on input conditioning. Specifically, unconditional image tokens are deterministically assigned to $N_u$ *unconditional experts*, each implemented as a feed-forward network (FFN), analogous to routed experts. Conversely, conditional image tokens are passed to the second step for fine-grained routing among routed experts. This explicit partitioning encourages experts to learn the functional disparity between token types, facilitating the specialization of unconditional and routed experts.

**Prototypical Routing.** The second step of our ProMoE router is to dispatch conditional image tokens for fine-grained expert allocation. Concretely, we introduce a novel prototypical routing mechanism where the routing weights are parameterized by a set of learnable prototypes $\mathbf{P} \in \mathbb{R}^{N_E \times D}$, as illustrated in Fig. 2. Each prototype $\mathbf{p}_i$ corresponds to an expert $E_i$ and is trained to represent the shared characteristics of a cluster of semantically similar tokens.

Compared with standard MoE token assignment, which computes pre-activation scores $\mathbf{Z} \in \mathbb{R}^{B \times L \times N_E}$ via a linear layer, we assign tokens using cosine similarity, which is more effective and naturally suited for measuring semantic similarity in latent space between tokens and prototypes:

$$\mathbf{Z}_{i,j} = [\mathcal{R}(\mathbf{X})]_{i,j} = \alpha \frac{\mathbf{x}_i \mathbf{p}_j^\top}{\|\mathbf{x}_i\| \|\mathbf{p}_j\|}, \tag{4}$$

where $\mathbf{x}_i$ and $\mathbf{p}_j$ are the $i$-th token in $\mathbf{X}$ and the $j$-th prototype in $\mathbf{P}$. $\alpha$ is a scaling factor.

Then, the activation function $\mathcal{A}$ transforms the pre-activation scores $\mathbf{Z}$ into token–expert affinity scores $\mathbf{S}$. Instead of softmax, which is computationally expensive and sensitive to sequence length, we opt for a simple monotonic function that preserves relative rankings. We evaluate both sigmoid and identity functions, finding that the identity $\mathcal{A}(\mathbf{Z}) = \mathbf{Z}$ performs best in practice, as shown in Table 8. We argue that the identity activation enables direct top-$K$ selection and provides stable training, thus improving performance. Consequently, we adopt identity activation as $\mathbf{S} = \mathbf{Z}$. Finally, each conditional image token is routed to the top-$K$ experts with gating scores $\mathbf{G}$, as in Eq. (3).

**Forward Process.** Besides unconditional and routed experts, we also incorporate $N_s$ *shared experts* that process all tokens to learn shared knowledge (Dai et al., 2024; Cheng et al., 2025). For each token, the output of our MoE block is defined as the sum of the shared experts' output and a selective output determined by the token type (conditional or unconditional):

$$\text{MoE}(\mathbf{x}) = \underbrace{\sum_{i=1}^{N_s} E_i^{\text{S}}(\mathbf{x})}_{\text{Shared}} + \begin{cases} \sum_{j=1}^{N_E} \mathbf{G}_j * E_j(\mathbf{x}) & \text{if } \mathbf{x} \in \mathbf{X}_c \\ \sum_{k=1}^{N_u} E_k^{\text{U}}(\mathbf{x}) & \text{if } \mathbf{x} \in \mathbf{X}_u \end{cases}, \tag{5}$$

where $E_i^{\text{S}}$, $E_j$, and $E_k^{\text{U}}$ are the shared, routed, and unconditional experts, respectively. $\mathbf{X}_c$ and $\mathbf{X}_u$ are conditional and unconditional image token sets, respectively, and $\mathbf{X}_c \cup \mathbf{X}_u = \mathbf{X}$.

To maintain a constant number of activated parameters, MoE models often employ fine-grained expert segmentation (Dai et al., 2024), where the inner hidden dimension of each expert is divided by the number of activated experts. In our most settings, each forward pass of our model activates exactly two experts: the single shared expert and one expert selected from the combined pool of routed and unconditional experts. Therefore, to match the computational cost of a dense model, we divide the hidden dimension of each expert's intermediate layer by a factor of two.

## 4.2 SEMANTIC ROUTING GUIDANCE

Due to the inherent high spatial redundancy of visual tokens, a naive MoE router fails to sufficiently distinguish tokens for effective routing, leading experts to learn homogeneous features. Consequently, additional semantic routing guidance is required to promote intra-expert coherence and inter-expert diversity. To validate this, we conduct experiments by augmenting the MoE router with two guidance types: *1)* Explicit Routing Guidance and *2)* Implicit Routing Guidance.

**Explicit Routing Guidance.** We design a routing classification loss that uses class labels to explicitly guide token assignment. Specifically, we manually partition the 1K ImageNet classes into $N_c$ superclasses based on coarse labels in (Feng & Patras, 2023), and allocate one expert per superclass. Since labels are sample-level, we instantiate the router as a classifier $\mathcal{C}$: we average-pool the input $\mathbf{X}$ over the token length dimension to obtain $\bar{\mathbf{X}}$, feed $\bar{\mathbf{X}}$ into $\mathcal{C}$ to produce sample–expert affinity scores $\bar{\mathbf{S}} \in \mathbb{R}^{B \times N_c}$, and assign the expert with highest score. During training, we supervise the routing process with a cross-entropy loss $\mathcal{L}_{\text{cls}} = \text{CE}(\bar{\mathbf{S}}, \bar{\mathbf{c}})$, where $\bar{\mathbf{c}}$ is the superclass label.

**Implicit Routing Guidance.** We replace the standard MoE router with k-means clustering, assigning all tokens in a cluster to a single expert. Unlike the routing classification loss that provides explicit supervision, this design offers implicit guidance by measuring token similarity, encouraging semantically similar tokens to be co-assigned. Concretely, we initialize $N_E$ cluster centroids by randomly sampling tokens. At each forward pass, we compute each token's distances to all centroids to obtain distance-based token–expert affinity scores. Each token is then assigned to its nearest centroid and thus routed to the corresponding expert. During training, centroids are updated iteratively by replacing each with the mean of their currently assigned tokens.

Results for both routing guidance are reported in Table 1, with all MoE models having the same activated parameters and comparable total parameters to ensure fairness. On the base model size, DiT-MoE (Fei et al., 2024) and DiffMoE (Shi et al., 2025) yield limited performance improvements. In contrast, adding either explicit or implicit guidance produces substantial gains. Notably, for both guidance strategies, we disable the load-balancing loss to isolate its routing effects; despite its importance for TC routing, guidance alone still markedly improves performance. These findings highlight the pivotal role of semantic routing guidance in vision MoEs.

Table 1: **Comparison results under Rectified Flow** on ImageNet ($256 \times 256$) after 500K training steps, evaluated with CFG=1.5.

| Model (500K) | FID50K $\downarrow$ | IS $\uparrow$ |
|---|---|---|
| Dense-DiT-B-Flow | 9.02 | 131.13 |
| DiT-MoE-B-Flow | 8.94 | 131.66 |
| DiffMoE-B-Flow | 8.22 | 137.46 |
| Classification-based Routing | 5.91 | 165.45 |
| K-Means-based Routing | 6.24 | 159.77 |

## 4.3 ENHANCING SEMANTIC ROUTING GUIDANCE VIA ROUTING CONTRASTIVE LEARNING

While the routing guidance strategies in Sec. 4.2 are effective, they suffer from key limitations: *1)* The classification-based routing loss is defined at the sample level, restricting token-level flexibility and requiring costly manual annotations, hindering generalization. *2)* Clustering-based routing supports only top-1 assignment, and struggles to scale to top-$K$, as methods like k-means rely on disjoint clusters, making multi-centroid assignment difficult. Moreover, k-means is sensitive to the number of clusters and the cluster initialization (Arthur & Vassilvitskii, 2006), reducing robustness.

To address these limitations, we propose the Routing Contrastive Loss (RCL), as illustrated in Fig. 2, to explicitly enhance semantic guidance in prototypical routing. Given a mini-batch of conditional image tokens, RCL encourages semantically similar tokens to be routed to the same expert and pushes dissimilar tokens toward different experts, prompting expert specialization in MoE. Concretely, for each prototype $\mathbf{p}_i$ associated with expert $E_i$, tokens assigned to $\mathbf{p}_i$ form the positive set, representing a cluster of semantically similar tokens, while tokens dispatched to other prototypes constitute the negative sets, comprising multiple clusters with semantics different from $\mathbf{p}_i$.

Next, RCL pulls each prototype $\mathbf{p}_i$ toward the centroid of its positive token set to enforce intra-expert coherence, while pushing it away from the centroids of negative sets to encourage inter-expert diversity. Let $\mathbf{X}_i$ denote the tokens assigned to expert $E_i$ in a mini-batch, its centroid $\mathbf{m}_i$ is computed as the token mean: $\mathbf{m}_i = \frac{1}{|\mathbf{X}_i|} \sum \mathbf{x} \in \mathbf{X}_i$. The RCL loss is then computed over the

Table 2: **Model configurations of ProMoE with different model sizes, aligning with DiT** (Peebles & Xie, 2023). "E14A1S1U1" denotes that a total of 14 experts are used, with 1 expert activated for each token, 1 expert shared by all tokens, and 1 unconditional expert for unconditional image tokens.

| Model Config | #Activated Params. | #Total Params. | #Experts | #Blocks L | #Hidden dim. D | #Head n |
|---|---|---|---|---|---|---|
| ProMoE-S | 33M | 75M | E14A1S1U1 | 12 | 384 | 6 |
| ProMoE-B | 130M | 300M | E14A1S1U1 | 12 | 768 | 12 |
| ProMoE-L | 458M | 1.063B | E14A1S1U1 | 24 | 1024 | 16 |
| ProMoE-XL | 675M | 1.568B | E14A1S1U1 | 28 | 1152 | 16 |

Table 3: **Quantitative comparison with Dense DiTs under Rectified Flow** on ImageNet (256×256) after 500K training steps, evaluated with CFG scales of 1.0 and 1.5.

| Model (500K) | # Activated Params. | # Total Params. | cfg=1.0 | | cfg=1.5 | |
|---|---|---|---|---|---|---|
| | | | FID50K ↓ | IS ↑ | FID50K ↓ | IS ↑ |
| Dense-DiT-B-Flow | 130M | 130M | 30.61 | 49.89 | 9.02 | 131.13 |
| ProMoE-B-Flow | 130M | 300M | **24.44** | **60.38** | **6.39** | **154.21** |
| Dense-DiT-L-Flow | 458M | 458M | 15.44 | 84.20 | 3.56 | 209.03 |
| ProMoE-L-Flow | 458M | 1.063B | **11.61** | **100.82** | **2.79** | **244.21** |
| Dense-DiT-XL-Flow | 675M | 675M | 13.38 | 91.57 | 3.23 | 227.05 |
| ProMoE-XL-Flow | 675M | 1.568B | **9.44** | **114.94** | **2.59** | **265.62** |

prototypes of $N_a$ experts that are assigned tokens in an online manner:

$$\mathcal{L}_{\text{RCL}} = -\frac{1}{N_a} \sum_{i=1}^{N_a} \log \frac{\exp(\text{sim}(\mathbf{p}_i, \mathbf{m}_i)/\tau)}{\sum_{j=1}^{N_a} \exp(\text{sim}(\mathbf{p}_i, \mathbf{m}_j)/\tau)}, \tag{6}$$

where $\text{sim}(\mathbf{a}, \mathbf{b}) = \frac{\mathbf{a}\mathbf{b}^\top}{\|\mathbf{a}\|\|\mathbf{b}\|}$ denotes cosine similarity, and $\tau$ is a temperature hyperparameter. Furthermore, we empirically find that the push-away operation in RCL acts as a load-balancing regularizer based on token semantics, and is more effective than traditional load-balancing loss (Shazeer et al., 2017) (see Appendix E.1). The final training loss of ProMoE is the combination of Eqs. (1) and (6), weighting $\mathcal{L}_{\text{RCL}}$ by a factor $\lambda_{\text{RCL}}$.

## 5 EXPERIMENT

### 5.1 EXPERIMENTAL SETUP

**Baseline and model configurations.** We compare against Dense-DiT (Peebles & Xie, 2023) and MoE baselines, including DiT-MoE (Fei et al., 2024), EC-DiT (Sun et al., 2024), and DiffMoE (Shi et al., 2025). For a fair comparison, all MoE models are evaluated with equivalent activated parameters to the dense model and comparable total parameters, training with both DDPM (Ho et al., 2020) and Rectified Flow (Esser et al., 2024a) objectives. We scale ProMoE across four sizes (S/B/L/XL) to align with established DiT benchmarks, as shown in Table 2. Models are named as: [Model]-[Size]-[Training Type], with an additional expert configuration. For instance, expert configuration E14A1S1U1 denotes 14 total experts (E14), top-1 activation (A1) over 12 routed experts, 1 shared expert (S1), and 1 unconditional expert (U1). More details are provided in Appendix A.

**Implementation details.** We conduct experiments on class-conditional image generation using the ImageNet (Deng et al., 2009) dataset, which contains 1,281,167 training images across 1,000 classes. Following (Peebles & Xie, 2023), we train all models with the AdamW optimizer with a learning rate of 1e-4. The batch size is 256, and weight decay is 0. We use horizontal flips as the only data augmentation, and a pretrained VAE from Stable Diffusion (Rombach et al., 2022b) to encode and decode images. We also maintain an exponential moving average (EMA) of model parameters during training with a decay rate of 0.9999, and all reported results use the EMA mode.

**Evaluation metrics.** We evaluate image generation quality of all methods using Fréchet Inception Distance (FID) (Heusel et al., 2017; Dhariwal & Nichol, 2021), calculated over 50K generated samples with 250 DDPM or Flow Matching Euler sampling steps. We also report Inception Score (IS) (Salimans et al., 2016) to measure the diversity of generated images.

Table 4: **Quantitative comparison with Dense DiTs under DDPM** on ImageNet (256×256) after 500K training steps, evaluated with CFG scales of 1.0 and 1.5.

| Model (500K) | # Activated Params. | # Total Params. | cfg=1.0 | | cfg=1.5 | |
|---|---|---|---|---|---|---|
| | | | FID50K ↓ | IS ↑ | FID50K ↓ | IS ↑ |
| Dense-DiT-B-DDPM | 130M | 130M | 41.19 | 35.94 | 18.61 | 78.71 |
| ProMoE-B-DDPM | 130M | 300M | **40.37** | **37.84** | **17.90** | **82.65** |
| Dense-DiT-L-DDPM | 458M | 458M | 20.81 | 65.51 | 6.29 | 148.38 |
| ProMoE-L-DDPM | 458M | 1.063B | **18.75** | **73.07** | **5.12** | **168.91** |
| Dense-DiT-XL-DDPM | 675M | 675M | 17.67 | 74.05 | 5.07 | 165.81 |
| ProMoE-XL-DDPM | 675M | 1.568B | **15.87** | **81.90** | **4.11** | **187.86** |

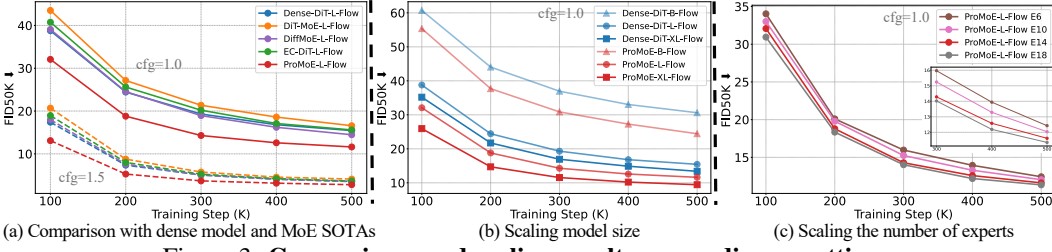

(a) Comparison with dense model and MoE SOTAs    (b) Scaling model size    (c) Scaling the number of experts

Figure 3: **Comparisons and scaling results across diverse settings.**

Table 5: **Quantitative comparison with MoE baselines under Rectified Flow** on ImageNet (256×256) after 500K training steps, evaluated with CFG scales of 1.0 and 1.5.

| Model (500K) | #Experts | # Activated Params. | # Total Params. | # GFLOPs | cfg=1.0 | | cfg=1.5 | |
|---|---|---|---|---|---|---|---|---|
| | | | | | FID50K ↓ | IS ↑ | FID50K ↓ | IS ↑ |
| DiT-MoE-L-Flow | E8A1S0U0 | 458M | 1.163B | 77.50 | 16.57 | 80.25 | 4.10 | 199.05 |
| EC-DiT-L-Flow | E8A1S0U0 | 458M | 1.163B | 77.50 | 15.58 | 84.11 | 3.65 | 209.06 |
| DiffMoE-L-Flow | E8A1S0U0 | 458M | 1.095B | 82.53 | 14.46 | 87.55 | 3.51 | 212.78 |
| DiffMoE-L-Flow | E16A1S0U0 | 458M | 1.846B | 90.03 | 13.55 | 92.33 | 3.30 | 222.40 |
| ProMoE-L-Flow | E14A1S1U1 | 458M | 1.063B | 77.72 | **11.61** | **100.82** | **2.79** | **244.21** |

## 5.2 MAIN RESULTS

**Comparison with Dense DiT.** The results in Fig. 3(a) and Tables 3 and 4 draw three conclusions: *1)* ProMoE consistently surpasses dense counterparts at equivalent activated parameters across all sizes, objectives, and CFG settings, demonstrating strong effectiveness, scalability, and generalization. *2)* Gains are more pronounced under Rectified Flow, the current dominant training paradigm, highlighting ProMoE's ability to scale modern diffusion models. Compared to dense models, without CFG, ProMoE-L-Flow reduces FID by 24.8% and increases IS by 19.7%; at the largest scale, ProMoE-XL-Flow reduces FID by 29.4%. With CFG=1.5, ProMoE-B-Flow reduces FID by 29.2% , while ProMoE-XL-Flow reduces FID by 19.8%. *3)* ProMoE is notably parameter-efficient; it uses fewer activated parameters yet outperforms dense models with more. Specifically, ProMoE-L-Flow achieves FID 11.61/2.79 at CFG 1.0/1.5, versus 13.38/3.23 for Dense-DiT-XL-Flow.

**Comparison with MoE SOTAs.** The results in Fig. 3(a), Tables 5 and 10 show that ProMoE outperforms all baselines across both objectives at equivalent activated parameters, with and without CFG. Without CFG, ProMoE-L-Flow reduces FID by 19.7% and increases IS by 15.2% relative to DiffMoE-L-Flow; with CFG=1.5, it reduces FID by 20.5% and increases IS by 14.8%. Notably, ProMoE-L-Flow (1.063B params) surpasses the larger DiffMoE-L-Flow with 16 experts (1.846B params), despite fewer total parameters, underscoring the effectiveness of our method.

**Comparison of computational cost and efficiency.** Tables 5 and 14 show that ProMoE achieves lower inference time and fewer GFLOPs than the SOTA MoE method DiffMoE, and maintains GFLOPs comparable to other MoE baselines while delivering substantially higher performance. This demonstrates that the performance gains primarily stem from our superior methodological design. More details about these experimental settings are provided in the Appendix A.

**Visualization Results.** Fig. 4 shows the samples generated by ProMoE-XL-Flow on ImageNet (256×256) after 2M training steps with CFG=4.0; see Appendix C.3 for more analyses and results.

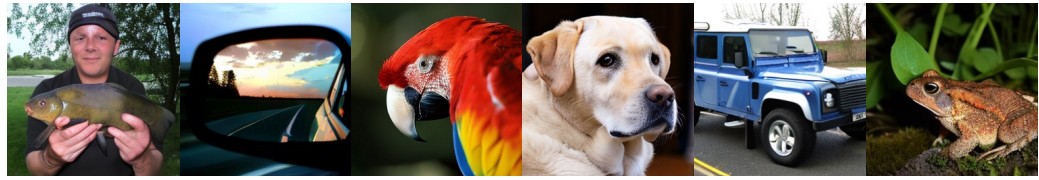

Figure 4: **Samples generated by ProMoE-XL-Flow** after 2M iterations with cfg=4.0.

Table 6: **Comparisons on GenEval benchmark under Rectified Flow** after 400K training steps.

| Model (400K) | #Experts | # Activated Params. | # Total Params. | GenEval ↑ | | | | | | |
|---|---|---|---|---|---|---|---|---|---|---|
| | | | | Single Obj. | Two Obj. | Counting | Colors | Position | Color Attr. | **Overall** |
| Dense Model | - | 3B | 3B | 0.840 | 0.275 | 0.362 | 0.611 | 0.095 | 0.155 | 0.390 |
| Token-Choice MoE | E5A1S0U0 | 3B | 12B | 0.856 | 0.320 | 0.334 | 0.627 | 0.157 | 0.207 | 0.417 |
| ProMoE | E5A1S0U1 | 3B | 12B | **0.884** | **0.371** | **0.418** | **0.675** | **0.212** | **0.217** | **0.463** |

**Comparison of training losses.** Fig. 5 shows training loss curves for our method, the dense model, and MoE baselines. At the L scale, ProMoE achieves lower loss and faster convergence than both MoE and dense baselines. This advantage persists at the largest XL scale, where

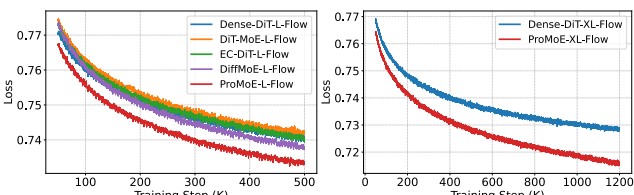

Figure 5: **Training loss curve comparisons.**

it continues to converge faster than the dense model, even with extended training up to 1.2M steps.

**Comparison on the text-to-image task.** We conduct text-to-image experiments to further demonstrate the generalization ability of ProMoE. The detailed experimental setup is provided in Appendix A, and we evaluate on the GenEval benchmark (Ghosh et al., 2023). As shown in Table 6, ProMoE significantly outperforms both the dense baseline and the Token-Choice MoE across the overall metric and all sub-tasks. These results strongly demonstrate ProMoE's robust generalization capabilities in challenging generation scenarios. Furthermore, we also provide qualitative results in Fig. 14, further verifying ProMoE's capability in high-quality text-to-image generation.

**Quantitative analysis of expert utilization.** We compare the expert utilization of ProMoE against the DiT-MoE baseline using averaged token-per-expert ratios computed on two disjoint class subsets, each consisting of 200 randomly sampled classes and 10,000 generated images. As shown in Fig. 6, DiT-MoE exhibits very similar token-per-expert distributions across different class subsets, and shows minimal variation in token proportions among experts within each subset, indicating poor specialization and a failure to induce diversity. In contrast, ProMoE demonstrates clear expert specialization, producing distinct utilization patterns across the disjoint subsets. Within each subset, experts exhibit varied usage frequencies without suffering from starvation or overuse. These results confirm that ProMoE achieves meaningful expert specialization and good load balancing.

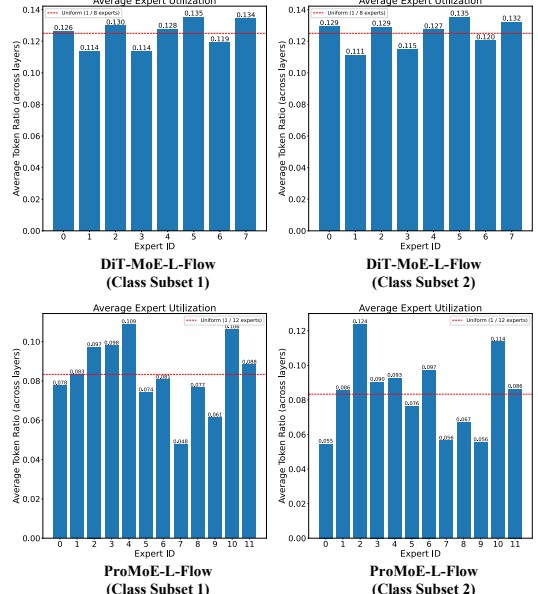

Figure 6: **Analysis of expert utilization.**

### 5.3 SCALING BEHAVIOR

**Scaling the model size.** As shown in Fig. 3(b), ProMoE exhibits consistent performance improvements over Dense-DiT when scaling from base (B) to large (L) to XL, with 130M, 458M, and 675M activated parameters, respectively, thereby validating the scalability of our method.

Table 7: **Ablation study of each component** on ImageNet (256×256) after 500K training steps, trained with Rectified Flow and evaluated with CFG scales of 1.0 and 1.5.

| Model (500K) | cfg=1.0 | | cfg=1.5 | |
|---|---|---|---|---|
| | FID50K ↓ | IS ↑ | FID50K ↓ | IS ↑ |
| Dense-DiT-B-Flow (Dense) | 30.61 | 49.89 | 9.02 | 131.13 |
| + Prototypical Routing | 27.93 | 53.35 | 7.92 | 140.86 |
| + Routing Contrastive Learning | 24.97 | 58.59 | 6.75 | 150.15 |
| + Conditional Routing | **24.44** | **60.38** | **6.39** | **154.21** |

Table 8: **Ablation of activation functions**, trained with Rectified Flow on ImageNet (256×256) for 500K steps.

| Activation (500K) | cfg=1.0 | | cfg=1.5 | |
|---|---|---|---|---|
| | FID↓ | IS↑ | FID↓ | IS↑ |
| Softmax | 25.74 | 58.04 | 6.92 | 149.11 |
| Sigmoid | 25.49 | 58.51 | 6.63 | 150.94 |
| Identity | **24.44** | **60.38** | **6.39** | **154.21** |

Table 9: **Ablation of conditional routing in K-Means-based Routing**, trained with Rectified Flow on ImageNet (256×256) for 500K steps.

| K-Means-based Routing (500K) | cfg=1.0 | | cfg=1.5 | |
|---|---|---|---|---|
| | FID↓ | IS↑ | FID↓ | IS↑ |
| w/o Cond. | 30.12 | 50.47 | 8.75 | 133.14 |
| w/ Cond. | **25.61** | **59.76** | **6.24** | **159.77** |

**Scaling the number of experts.** Fig. 3(c) shows monotonic gains as the expert number increases from 4 to 16, with 1 shared and 1 unconditional expert per setup. For a fair comparison, we maintain comparable total parameters over MoE baselines and use 14 experts across settings.

## 5.4 ABLATION STUDIES

**Ablation on each component.** Table 7 shows that using prototypical routing improves performance and surpasses DiT-MoE-B-Flow and DiffMoE-B-Flow (see Table 1). Adding routing contrastive learning (RCL) yields substantial gains, reducing FID by 10.6% and increasing IS by 9.8%, highlighting the importance of semantic routing guidance. These results validate the effectiveness of each component.

**Ablation on score activation function.** Since our prototypical routing computes similarities in latent space, choosing an appropriate activation to map similarities into routing scores is crucial. As shown in Table 8, the identity mapping yields the best performance, sigmoid is second-best, and softmax performs worst. Consequently, we adopt the identity function as the score activation.

**Ablation on conditional routing within the K-Means–based Routing.** We emphasize that the proposed conditional routing is a general, method-agnostic component that can benefit other routing schemes. We ablate it within the K-Means–based Routing method in Sec. 4.2. As shown in Table 9, removing conditional routing significantly degrades performance, underscoring its importance.

## 6 CONCLUSION

In this paper, we present ProMoE, a Mixture-of-Experts framework featuring a two-step router with explicit routing guidance to promote expert specialization. We analyze differences between language and vision tokens: discrete text tokens are semantically dense, whereas visual tokens exhibit high spatial redundancy and functional heterogeneity, hindering the effectiveness of MoE in DiT models. To address this, we introduce routing guidance based on the token's functional role and semantic content, yielding a two-step router comprising conditional routing and prototypical routing. Furthermore, we propose a routing contrastive loss that enhances semantic guidance in prototypical routing, explicitly promoting intra-expert coherence and inter-expert diversity. Extensive experiments demonstrate that ProMoE outperforms dense DiT and existing MoE SOTAs, even with fewer activated or total parameters, providing a robust solution for applying MoE to DiT models.

**Limitations.** While we follow standard evaluation protocols and report FID50K and IS, these metrics may not fully capture fine-grained perceptual quality or semantic faithfulness.

**Acknowledgements** This work was supported in part by National Natural Science Foundation of China (No. 62471148), STI2030-Major Projects (No. 2021ZD0200204), Shanghai Center for Brain Science and Brain-inspired Technology, and Alibaba Group.

## ETHICS STATEMENT

Our method achieves substantial improvements on the ImageNet benchmark over dense DiT and state-of-the-art MoE methods, providing an effective solution for scaling DiT with MoE. Nonetheless, it inherits common risks of generative models, such as the potential to create fake data. Robust image forgery detection may help mitigate these concerns. In addition, we adhere to ethical guidelines in all experiments.

## REPRODUCIBILITY STATEMENT

We make the following efforts to ensure the reproducibility of ProMoE: (1) All experiments are conducted on the publicly available ImageNet-1K benchmark. (2) Our code and trained model weights will be made publicly available. (3) We provide implementation details in Sec. 5.1 and Appendix A.

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

APPENDIX

## A    EXPERIMENTAL SETUP

**Baselines.**    We compare against open-source state-of-the-art DiT-based MoE methods, with activated parameters equivalent to the dense model and comparable total parameter counts. We implement DiT-MoE (Fei et al., 2024), EC-DiT (Sun et al., 2024), and DiffMoE (Shi et al., 2025) following their original papers and referring to the open-source repository[1]. All methods are trained with the same training settings, including learning rate, batch size, and data augmentation.

**Implementation details.**    We train ProMoE with two objectives: the standard DDPM objective (Ho et al., 2020), and Rectified Flow with Logit-Normal sampling from SD3 (Esser et al., 2024a) to align better with modern DiT training paradigms (*e.g.*, Sana (Xie et al., 2024)). Besides the hyperparameters in Sec. 5.1, we provide additional details as follows. In prototypical routing, $\alpha$ in Eq. (4) is set to 1. For routing contrastive learning, the temperature is set to 0.07, and the loss weight $\lambda_{\text{RCL}}$ is 1 unless stated otherwise.

**Evaluation metrics.**    We follow the standard DiT evaluation protocol (Peebles & Xie, 2023), computing FID and IS on 50,000 generated images[2] at classifier-free guidance scales of 1.0 and 1.5.

**Computational cost evaluation settings.**    We report the computational cost of our model in terms of training time, inference time, and FLOPs, and compare it against dense and MoE baselines. The inference time is measured for the core denoising process (excluding VAE decoding) with a batch size of 128, a classifier-free guidance (CFG) scale of 1.5, and 250 sampling steps. For a fair comparison, all experiments are conducted on 4 NVIDIA A800 GPUs under identical hardware settings.

**Text-to-image experimental setup.**    We conduct new experiments on the text-to-image generation task to demonstrate the generalization of our ProMoE. Below, we detail the experimental setup. **1)** For the model architecture, we adopt an MM-DiT–style architecture, where text and visual tokens are concatenated and jointly processed via self-attention for effective conditioning. The dense baseline uses 36 layers, 16 attention heads, a hidden size of 11,008, and has 3B parameters. We build on this backbone by extending it to an MoE architecture, where each expert is a visual FFN. As a comparison baseline, we adopt the standard MoE paradigm commonly used in LLMs, denoted as Token-Choice MoE, which uses 5 routed experts and has 12B total parameters with 3B activated parameters. Our ProMoE uses the same overall capacity (12B total, 3B activated parameters) and also employs 5 experts (1 unconditional and 4 routed), ensuring a fair comparison under matched parameter and activation budgets. **2)** For implementation details, we train all models on a 2M-image subset of LAION-5B (Schuhmann et al., 2022) to enable rapid validation. We use the AdamW optimizer with a learning rate of 1e-4, a global batch size of 384, and a weight decay of 0.02. Images are encoded into the latent space using the pretrained Wan2.5-Preview VAE (Wan et al., 2025), and text is encoded with Qwen2.5-VL (Bai et al., 2025). The training objective is Rectified Flow. Both the baselines and our ProMoE are trained under the same settings to ensure a fair comparison. All experiments are conducted on 16 A800 GPUs for 400K training steps. For inference, we use the EMA model and adopt the UniPC (Zhao et al., 2023) sampler with 50 sampling steps, with a classifier-free guidance scale of 5.0.

## B    IMPLEMENTATION ALGORITHMS

The implementation algorithm of ProMoE is provided in Algorithm 1. In the algorithm, input class labels are used solely to distinguish conditional image tokens from unconditional image tokens. During inference, if classifier-free guidance (CFG) is disabled, all tokens are dispatched to the routed experts and the shared expert; the unconditional expert is not used. If CFG is enabled, class labels are replaced with a batch-level binary mask indicating which samples receive conditioning (*i.e.*, treated as conditional image tokens). No routing contrastive loss is computed during inference.

---

[1] https://github.com/KwaiVGI/DiffMoE
[2] https://github.com/openai/guided-diffusion/tree/main/evaluations

## C  MORE RESULTS

### C.1  MORE t-SNE VISUALIZATIONS OF LANGUAGE AND VISUAL TOKENS

To further validate the findings on differences between language and visual tokens in Sec. 1, we extend the visualization results in Fig. 1(a). Figs. 11 and 12 present t-SNE visualizations of token embeddings from DiT-XL/2 (Peebles & Xie, 2023) and Llama-3 8B (Dubey et al., 2024) across different layers; for DiT-XL/2, we also visualize tokens at different diffusion timesteps. To facilitate comparison, we cluster token embeddings into 10 groups using k-means. For model inputs, we randomly sample 110 ImageNet classes, feed the corresponding class labels to DiT-XL/2 and the class names to Llama-3 8B, and randomly select 1K intermediate-layer tokens for visualization. The results in Figs. 11 and 12 further confirm that language tokens are semantically dense with high inter-token differences, whereas visual tokens exhibit high spatial redundancy.

### C.2  t-SNE VISUALIZATIONS OF TOKEN ASSIGNMENTS

To assess the impact of visual-token redundancy on MoE expert selection, as indicated by Fig. 1(a) and Figs. 11 and 12, we visualize intermediate-layer token assignments of ProMoE-L-Flow and DiT-MoE-L-Flow at 500K training steps without classifier-free guidance, as shown in Fig. 7. Following Sec. C.1, we randomly sample 110 ImageNet classes, feed the corresponding class labels to both ProMoE and DiT-MoE, and randomly select 2,560 tokens from an intermediate-layer MoE block to visualize the expert selection of each token. Compared with token-choice MoE methods such as DiT-MoE, our approach assigns experts according to token semantics, producing well-formed clusters in the token-embedding space: semantically similar tokens form compact clusters and are routed to the same expert, whereas clusters assigned to different experts are clearly separated. These results further corroborate the importance of explicit routing guidance for visual MoE, and our method achieves effective intra-expert coherence and inter-expert diversity.

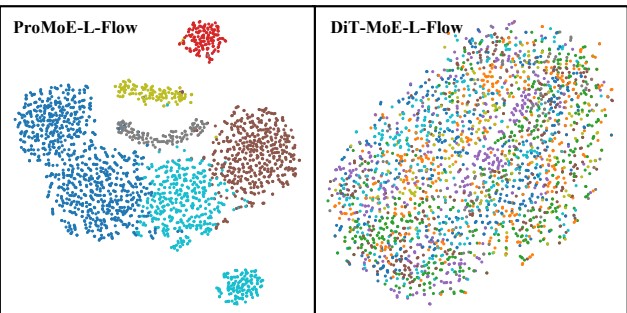

Figure 7: **t-SNE visualization results of ProMoE and DiT-MoE on expert allocation (token assignment).** Each color corresponds to a single expert.

### C.3  MORE VISUALIZATION RESULTS

We provide additional generation results in Figs. 13 and 14. Our method produces high-quality images across both simple and challenging categories.

### C.4  MORE COMPARISON RESULTS USING THE DDPM OBJECTIVE

We provide additional comparisons with dense models and MoE SOTAs. Besides the FID results in Fig. 3(a), we also report Inception Score comparisons, as shown in Fig. 8. In addition, we present quantitative comparisons with MoE SOTAs under the DDPM objective in Table 10. Across both training objectives and CFG settings, our method consistently outperforms the dense model and existing MoE SOTAs, demonstrating its effectiveness.

Table 10: **Quantitative comparison with MoE baselines under DDPM** on ImageNet (256×256) after 500K training steps, and evaluated at CFG scales of 1.0 and 1.5.

| Model (500K) | #Experts | # Activated Params. | # Total Params. | cfg=1.0 | | cfg=1.5 | |
|---|---|---|---|---|---|---|---|
| | | | | FID50K ↓ | IS ↑ | FID50K ↓ | IS ↑ |
| DiT-MoE-L-DDPM | E8A1S0U0 | 458M | 1.163B | 23.12 | 60.08 | 7.55 | 133.63 |
| EC-DiT-L-DDPM | E8A1S0U0 | 458M | 1.163B | 20.76 | 64.77 | 6.48 | 146.49 |
| DiffMoE-L-DDPM | E8A1S0U0 | 458M | 1.095B | 19.45 | 70.93 | 5.47 | 158.30 |
| ProMoE-L-DDPM | E14A1S1U1 | 458M | 1.063B | **18.75** | **73.07** | **5.12** | **168.91** |

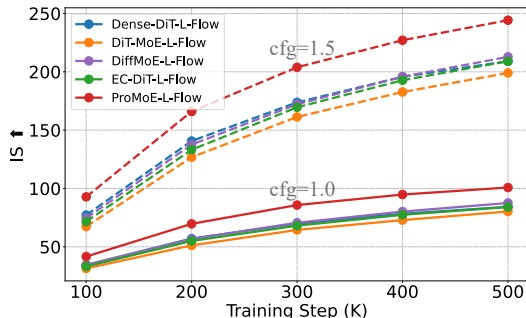

Figure 8: **Comparison with dense model and MoE SOTAs on Inception Score.**

## C.5 MORE COMPARISON RESULTS WITH EXTENDED TRAINING STEPS

We provide comparison results between our ProMoE and Dense DiT on more training steps in Tables 11 and 12. We observe that ProMoE-L-Flow at 500K steps surpasses Dense-DiT-L-Flow at 1M steps on FID, and ProMoE-XL-Flow at 500K steps surpasses Dense-DiT-XL-Flow at 1M steps on both FID and IS. With longer training, ProMoE-L-Flow at 1M steps outperforms Dense-DiT-L-Flow at 2M steps and Dense-DiT-XL-Flow at 1M steps. These findings are consistent with those in Sec. 5.2, demonstrating faster convergence and scalability of our method.

We also evaluate L and XL ProMoE models against dense and SOTA DiffMoE baselines at 2M steps with CFG scales of 1.0, 1.5, and 4.0. As shown in Table 12, our ProMoE models consistently outperform baselines for both L/XL sizes under various settings. At CFG=4.0, for L-size, ProMoE reduces FID by ∼21% over the dense model and 10% over DiffMoE; for XL-size, ProMoE reduces FID by ∼30% over the dense model and 16% over DiffMoE, while also achieving the best IS. At lower CFG values, ProMoE still yields consistent improvements over both dense and DiffMoE baselines. Furthermore, ProMoE-L at 2M steps surpasses Dense/DiffMoE-XL on FID with fewer activated parameters at CFG scales of 4.0 and 1.5. These results further solidify ProMoE's better performance and parameter efficiency under prolonged training and stronger guidance.

Table 11: **Quantitative comparison with Dense DiTs under Rectified Flow** on ImageNet (256×256) after more training steps, evaluated with CFG scales of 1.0 and 1.5.

| Model (Training Steps) | # Activated Params. | # Total Params. | cfg=1.0 | | cfg=1.5 | |
|---|---|---|---|---|---|---|
| | | | FID50K ↓ | IS ↑ | FID50K ↓ | IS ↑ |
| Dense-DiT-L-Flow (1M) | 458M | 458M | 12.21 | 100.97 | 2.97 | 245.63 |
| ProMoE-L-Flow (500K) | 458M | 1.063B | 11.61 | 100.82 | 2.79 | 244.21 |
| ProMoE-L-Flow (1M) | 458M | 1.063B | **9.88** | **118.91** | **2.75** | **278.22** |
| Dense-DiT-XL-Flow (1M) | 675M | 675M | 10.67 | 107.68 | 2.82 | 260.61 |
| ProMoE-XL-Flow (500K) | 675M | 1.568B | 9.44 | 114.94 | 2.59 | 265.62 |
| ProMoE-XL-Flow (1M) | 675M | 1.568B | **8.34** | **128.58** | **2.53** | **292.38** |

## C.6 INCREASING THE NUMBER OF ACTIVATED EXPERTS

As discussed in Sec. 4.2, classification- and clustering-based routing inherently do not support top-k assignment, permitting only top-1. In contrast, ProMoE is more flexible and scalable, and supports

Table 12: **Quantitative comparison under Rectified Flow after 2M training steps** on ImageNet (256×256), evaluated with CFG scales of 1.0, 1.5, and 4.0.

| Model (Training Steps) | # Activated Params. | # Total Params. | cfg=1.0 | | cfg=1.5 | | cfg=4.0 | |
|---|---|---|---|---|---|---|---|---|
| | | | FID50K ↓ | IS ↑ | FID50K ↓ | IS ↑ | FID50K ↓ | IS ↑ |
| Dense-DiT-L-Flow (2M) | 458M | 458M | 10.55 | 112.55 | 2.81 | 266.24 | 17.71 | 464.25 |
| DiffMoE-L-Flow (2M) | 458M | 1.163B | 10.09 | 122.01 | 2.57 | 276.75 | 15.70 | 465.60 |
| ProMoE-L-Flow (2M) | 458M | 1.063B | **9.67** | **125.88** | **2.22** | **290.61** | **14.05** | **466.96** |
| Dense-DiT-XL-Flow (2M) | 675M | 675M | 9.30 | 121.92 | 2.55 | 282.59 | 17.05 | 474.39 |
| DiffMoE-XL-Flow (2M) | 675M | 1.735B | 8.83 | 134.37 | 2.49 | 297.25 | 14.25 | 475.13 |
| ProMoE-XL-Flow (2M) | 675M | 1.568B | **8.16** | **139.66** | **2.08** | **304.26** | **11.95** | **478.60** |

top-k assignment. To validate this, we increase the number of activated routed experts from 1 to 3, which raises the activated parameters while keeping the total parameter count unchanged. As shown in Table 13, this increase yields improved performance, confirming the effectiveness and scalability of our method.

Table 13: **Results of increasing the number of activated routed experts** on ImageNet (256×256) after 500K steps, trained with Rectified Flow and evaluated at CFG scales 1.0 and 1.5.

| Model (500K) | #Experts | # Activated Params. | # Total Params. | cfg=1.0 | | cfg=1.5 | |
|---|---|---|---|---|---|---|---|
| | | | | FID50K ↓ | IS ↑ | FID50K ↓ | IS ↑ |
| ProMoE-L-Flow | E14A1S1U1 | 458M | 1.063B | 11.61 | 100.82 | 2.79 | 244.21 |
| ProMoE-L-Flow | E14A3S1U1 | 558M | 1.063B | **11.40** | **103.78** | **2.72** | **246.96** |

## C.7 FULL DETAILS OF COMPUTATIONAL COST AND EFFICIENCY

We provide a comprehensive analysis of computational costs, including detailed comparisons of training time, inference time, and GFLOPs. These results verify the inherent efficiency of our approach and demonstrate that our significant performance gains are primarily attributable to superior methodological design rather than increased computational overhead.

Table 14: **Comparison of Training Time, Inference Time, and FLOPs with Dense Model and MoE baselines under Rectified Flow** on ImageNet (256×256) after 500K training steps.

| Model (500K) | #Experts | # Activated Params. | # Total Params. | # Training Time | # Inference Time (cfg=1.5) | # GFLOPs |
|---|---|---|---|---|---|---|
| Dense-DiT-L-Flow | - | 458M | 458M | 166.67 GPU Hours | 1.25 sec/sample | 77.50 |
| DiT-MoE-L-Flow | E8A1S0U0 | 458M | 1.163B | 333.33 GPU Hours | 1.49 sec/sample | 77.50 |
| EC-DiT-L-Flow | E8A1S0U0 | 458M | 1.163B | 277.78 GPU Hours | 1.49 sec/sample | 77.50 |
| DiffMoE-L-Flow | E8A1S0U0 | 458M | 1.095B | 333.33 GPU Hours | 1.64 sec/sample | 82.53 |
| ProMoE-L-Flow | E14A1S1U1 | 458M | 1.063B | 333.33 GPU Hours | 1.53 sec/sample | 77.72 |

## C.8 COMPARISON WITH OTHER UNSUPERVISED CLUSTERING METHODS

We conduct additional experiments comparing our ProMoE with a GMM-based routing baseline on the large (L) model. The results in Table 15 show that ProMoE consistently outperforms the GMM-based routing baseline across all metrics (FID and IS) and CFG scales under both top-1 (E14A1) and top-3 (E14A3) activation settings. We argue that GMM, like other clustering-based routing schemes, relies purely on implicit learning and lacks explicit guidance, which is insufficient for robust expert specialization in vision MoE, especially for large-scale models. In contrast, ProMoE achieves better performance, highlighting the robustness and effectiveness of our explicit semantic routing guidance strategies.

Table 15: **Quantitative comparison with GMM-based routing baseline under Rectified Flow** on ImageNet (256×256) after 500K training steps, evaluated with CFG scales of 1.0 and 1.5.

| Model (500K) | #Experts | # Activated Params. | # Total Params. | cfg=1.0 | | cfg=1.5 | |
|---|---|---|---|---|---|---|---|
| | | | | FID50K ↓ | IS ↑ | FID50K ↓ | IS ↑ |
| GMM-based Routing | E14A1S1U1 | 458M | 1.063B | 15.56 | 84.94 | 3.76 | 206.05 |
| ProMoE-L-Flow | E14A1S1U1 | 458M | 1.063B | **11.61** | **100.82** | **2.79** | **244.21** |
| GMM-based Routing | E14A3S1U1 | 558M | 1.063B | 15.44 | 86.09 | 3.72 | 208.15 |
| ProMoE-L-Flow | E14A3S1U1 | 558M | 1.063B | **11.40** | **103.78** | **2.72** | **246.96** |

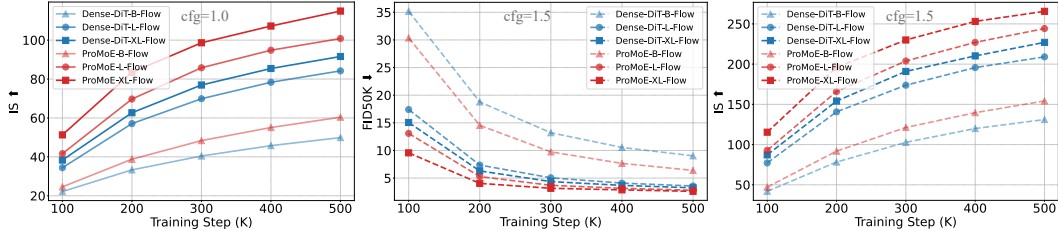

Figure 9: **More scaling results on model size.**

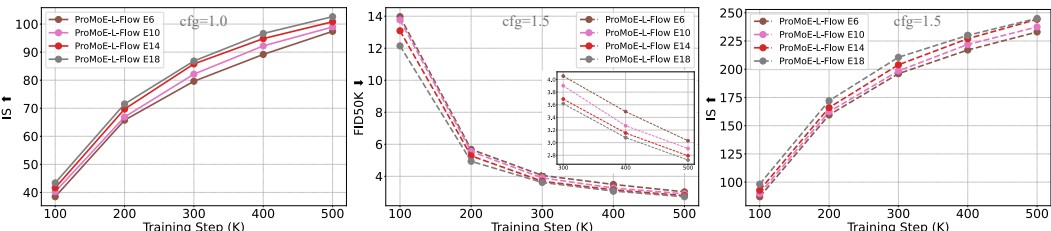

Figure 10: **More scaling results on the number of experts.**

## D  MORE RESULTS ON SCALING BEHAVIOR

### D.1  SCALING MODEL SIZE

Fig. 3(b) shows FID results for model size scaling at CFG=1.0. We additionally report FID results at CFG=1.5 and Inception Score at CFG=1.0 and 1.5, as shown in Fig. 9. ProMoE consistently outperforms its dense counterparts, and ProMoE-L-Flow surpasses Dense-XL-Flow in terms of FID and Inception Score at both CFG=1.0 and 1.5, despite using fewer activated parameters. These observations are consistent with those in Sec. 5.2.

### D.2  SCALING THE NUMBER OF EXPERTS

We report Inception Score for scaling the number of experts at CFG=1.0 and 1.5, and FID at CFG=1.5, as shown in Fig. 10. Performance of ProMoE improves as the number of experts increases, demonstrating the scalability of our approach.

### D.3  SCALING THE NUMBER OF EXPERTS ON K-MEANS-BASED ROUTING

We conduct an additional experiment varying the expert number for K-means-based routing (unconditional routing is not used here). As shown in Table 16, K-means-based routing is highly sensitive to the expert number and fails to scale effectively: increasing experts yields fluctuating performance with no clear gain, despite the increased parameter count. This contrast further highlights the scalability and robustness of our ProMoE compared to clustering-based routing.

Table 16: **Ablation of expert number in K-Means-based Routing on base model size**, trained with Rectified Flow on ImageNet (256×256) for 500K steps, evaluated with CFG=1.5. The setup uses 1 activated expert and 1 shared expert.

| Number of Experts | 3 | 5 | 9 | 13 | 21 |
|---|---|---|---|---|---|
| FID 10K↓ | 12.55 | 11.52 | 12.43 | 11.41 | 12.15 |

# E   MORE ABLATION STUDIES

## E.1   ABLATION ON LOAD-BALANCING LOSS

As discussed in Sec. 4.3, the push-away term in our routing contrastive learning (RCL) serves a role similar to load balancing. We verify this with an ablation in Table 17. Adding a conventional load-balancing loss on top of our method slightly degrades performance. We attribute this to RCL's explicit semantic guidance: it leverages token semantics to maintain diverse expert assignments, whereas load balancing loss only regularizes token counts and ignores assignment quality and semantics, thereby interfering with RCL. These results indicate that the semantic routing guidance from RCL is more effective than traditional load-balancing losses.

Table 17: **Ablation study of using load-balancing loss under Rectified Flow** on ImageNet (256×256) after 500K training steps.

| Model (500K) | cfg=1.0 | | cfg=1.5 | |
|---|---|---|---|---|
| | FID50K ↓ | IS ↑ | FID50K ↓ | IS ↑ |
| w/ load-balancing loss | 24.98 | 59.04 | 6.53 | 151.37 |
| w/o load-balancing loss | **24.44** | **60.38** | **6.39** | **154.21** |

## E.2   ABLATION ON LOSS WEIGHT OF ROUTING CONTRASTIVE LEARNING

We vary the loss weight of routing contrastive learning (RCL) and report the results in Table 18. We observe that RCL is insensitive to the loss weight, as increasing it from 1 to 10 yields only marginal gains. Therefore, we use a default weight of 1 for all experiments, except for ProMoE-B-DDPM, which uses a weight of 10 based on this ablation study.

Table 18: **Ablation study of $\lambda_{RCL}$ in ProMoE-B-DDPM** on ImageNet (256×256) after 500K training steps.

| $\lambda_{RCL}$ (500K) | cfg=1.0 | | cfg=1.5 | |
|---|---|---|---|---|
| | FID50K ↓ | IS ↑ | FID50K ↓ | IS ↑ |
| 1 | 40.48 | 36.77 | 18.34 | 80.07 |
| 2 | 40.37 | 37.46 | 18.01 | 81.88 |
| 5 | **40.33** | 37.08 | 18.03 | 81.1 |
| 10 | 40.37 | **37.84** | **17.90** | **82.65** |

# F   USAGE OF LARGE LANGUAGE MODELS (LLMS)

In accordance with the ICLR 2026 policy, we report our use of a large language model (LLM) in preparing this manuscript. The LLM's role was strictly confined to language polishing, such as correcting grammar, refining wording, and improving readability. All scientific contributions, including the ideation, methodology, experimental design, and final conclusions, are entirely our own. The LLM was used solely as a writing-enhancement tool and did not contribute to the scientific aspects of the work. We have reviewed the manuscript and take full responsibility for its content.

---

**Algorithm 1** `ProMoE` Layer (Training)

---

**Input:** $\mathbf{X} \in \mathbb{R}^{B \times L \times D}$ (input sequence), $\mathbf{c} \in \mathbb{Z}^B$ (batch labels)
**Variables:** $N_E$ (number of routed experts), $K$ (number of activated routed experts), $\mathbf{P} \in \mathbb{R}^{N_E \times D}$ (Learnable prototypes for routing), $E$ (List of routed expert FFNs), $E^{\mathrm{U}}$ (Unconditional expert FFN), $E^{\mathrm{S}}$ (Shared expert FFN), $\lambda_{\mathrm{RCL}}$ (coef of Routing contrastive loss), $\tau$ (temperature)

1: **Initialize:** $\mathbf{O} \leftarrow$ zeros_like($\mathbf{X}$)            ▷ Initialize final output
2: /*** Step 1. Functional Routing ***/
3:    $M_u \leftarrow (\mathbf{c} ==$ empty conditioning$)$          ▷ Get mask of unconditional image tokens
4:    $M_c \leftarrow \neg M_u$               ▷ Get mask of conditional image tokens
5:    $\mathbf{X}_u \leftarrow \mathbf{X}[\text{expand}(M_u)]$          ▷ Get unconditional image tokens
6:    $\mathbf{X}_c \leftarrow \mathbf{X}[\text{expand}(M_c)]$            ▷ Get conditional image tokens
7: /*** Step 2. Unconditional Image Tokens Processing ***/
8:    $\mathbf{O}_{\mathrm{U}} \leftarrow E^{\mathrm{U}}(\mathbf{X}_u)$
9:    $\mathbf{O}[M_u] \leftarrow \mathbf{O}_{\mathrm{U}}$
10: **if** any$(M_c)$ **then**
11:      /*** Step 3. Prototypical Routing ***/
12:      $\mathbf{X}'_c \leftarrow$ reshape$(\mathbf{X}_c, (-1, D))$         ▷ Flatten conditional image tokens
13:      $n_c \leftarrow \mathbf{X}'_c$.shape[0]           ▷ Get number of conditional image tokens
14:      $\mathbf{Z} \in \mathbb{R}^{n_c \times N_E} \leftarrow$ L2_Normalize$(\mathbf{X}'_c) \times$ L2_Normalize$(\mathbf{P})^{\top}$    ▷ Get pre-activation scores
15:      $\mathbf{S} \leftarrow$ Identity$(\mathbf{Z})$            ▷ Get token–expert affinity scores
16:      $\mathbf{G} \in \mathbb{R}^{n_c \times K}$, indices $\in \mathbb{Z}^{n_c \times K} \leftarrow$ TopK$(\mathbf{S}, K)$     ▷ Get gating tensor and indices
17:      /*** Step 4. Conditional Image Tokens Processing ***/
18:      $\mathbf{O}'_{\mathrm{C}} \leftarrow$ zeros_like$(\mathbf{X}'_c)$
19:      **for** $i \leftarrow 0$ **to** $N_E - 1$ **do**
20:         $m_i \leftarrow$ (indices $== i$).any$(\text{dim} = 1)$      ▷ Mask of tokens routed to expert $i$
21:         **if** any$(m_i)$ **then**
22:            $\mathbf{G}_i \leftarrow$ sum$(\mathbf{G}[m_i] \times (\text{indices}[m_i] == i), \text{dim} = 1)$      ▷ Final gating scores
23:            $\mathbf{O}'_{\mathrm{C}}[m_i] \leftarrow \mathbf{O}'_{\mathrm{C}}[m_i] + \mathbf{G}_i$.unsqueeze$(1) \times E_i(\mathbf{X}'_c[m_i])$    ▷ Update final output
24:         **end if**
25:      **end for**
26:      $\mathbf{O}^{\mathrm{C}} \leftarrow$ reshape$(\mathbf{O}'_{\mathrm{C}}, \mathbf{X}_c$.shape$)$
27:      $\mathbf{O}[M_c] \leftarrow \mathbf{O}[M_c] + \mathbf{O}^{\mathrm{C}}$
28:      /*** Step 5. Routing Contrastive Learning ***/
29:      aux_loss $\leftarrow \lambda_{\mathrm{RCL}} \times \mathcal{L}_{\mathrm{RCL}}(\mathbf{X}_c, \text{indices}, \mathbf{P}, \tau)$
30: **end if**
31: /*** Step 6. Shared Expert Processing ***/
32: $\mathbf{O} \leftarrow \mathbf{O} + E^{\mathrm{S}}(\mathbf{X})$
33: **Return:** $\mathbf{O}$, aux_loss

---

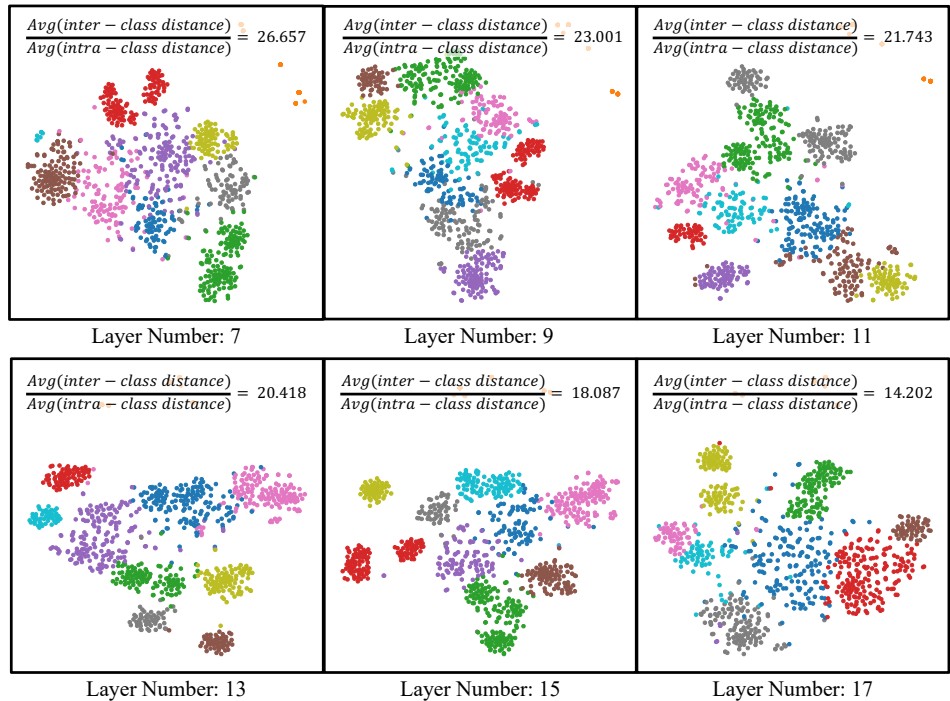

Figure 11: **More t-SNE visualization results of Llama-3 8B** on different layers.

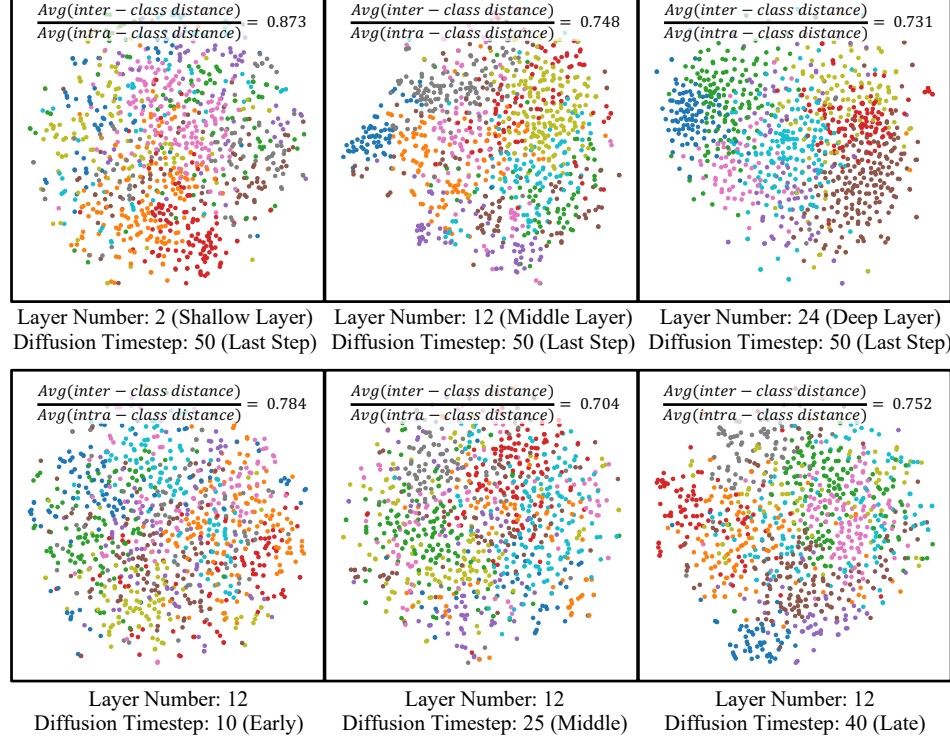

Figure 12: **More t-SNE visualization results of DiT-XL/2** on different layers and diffusion timesteps.

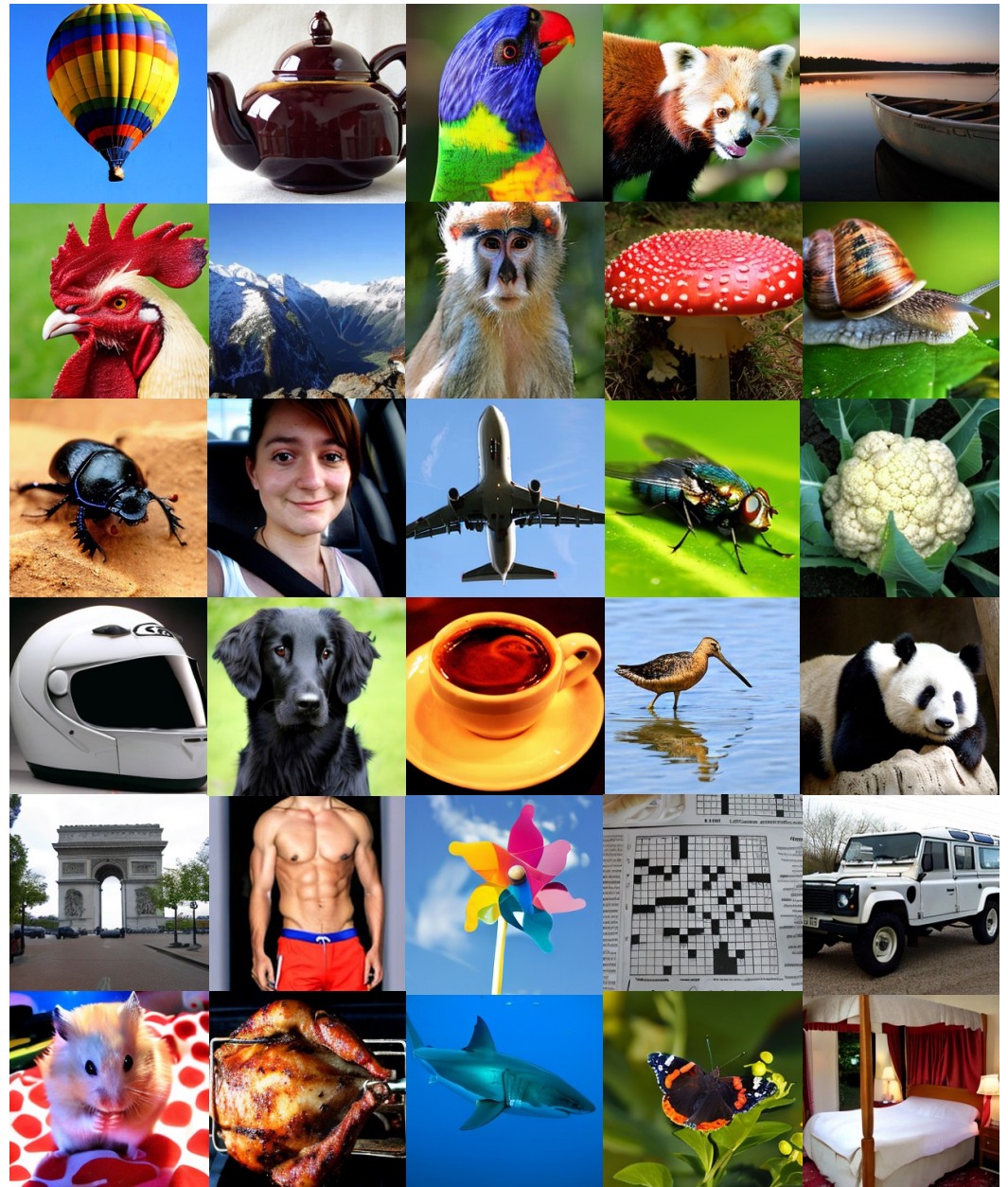

Figure 13: **More samples generated by ProMoE-XL-Flow** after 2M iterations with cfg=4.0.

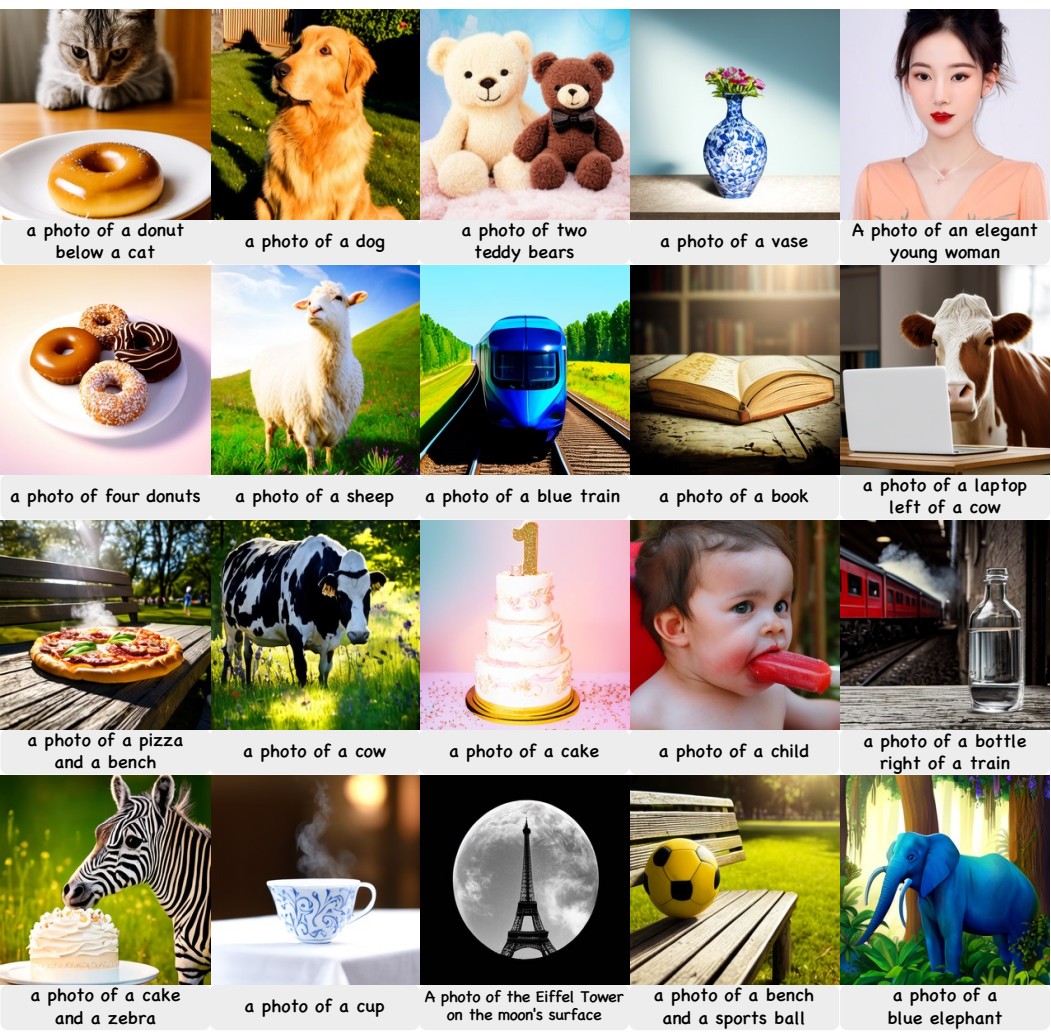

Figure 14: **Samples generated by ProMoE on the Text-to-Image task**.

