# OpenReview forum: "Routing Matters in MoE: Scaling Diffusion Transformers with Explicit Routing Guidance"
_ICLR.cc/2026/Conference — ICLR 2026 Poster_

### Official Review · Reviewer_vTUm · 2025-10-25

**Soundness:** 3
**Presentation:** 2
**Contribution:** 2
**Rating:** 2
**Confidence:** 3

**Summary:**

The authors present ProMoE, a two-step routing framework for Diffusion-MoE models. In the first step, tokens are routed according to functional heterogeneity, sending unconditional tokens to dedicated unconditional experts. The remaining conditional tokens are then routed using a cosine similarity router to experts, with an additional routing contrastive loss designed to enhance intra-expert semantic similarity and inter-expert semantic differences.

**Strengths:**

1. Experimental results on ImageNet are strong, with uniform improvements in generation quality and diversity

2. The authors address scalability comprehensively, with experiments covering four model sizes.

3. The paper is clear and easy to follow

**Weaknesses:**

**Limited technical novelty**. The authors claim the prototypical routing mechanism is novel, but I struggle to see the novelty here. The learnable prototypes appear to just be standard learnable expert embeddings and semantic similarity is computed using a cosine similarity between inputs and expert embeddings, which is fairly conventional in MoE [1,2, 3]. The authors do use an identity activation function which is non-standard, but this alone does not really qualify the entire routing mechanism as novel in my view.

**Missing experimental results**. The reported results in Fig 3 show that ProMoE indeed offers substantive improvements over dense and MoE baselines up to 500k training steps, but the performance differences do appear to be converging on one another very quickly, with the differences starting to look more marginal towards 500K and with increasing cfg to 1.5. Given that the samples generated in figure 4 required 2 million training steps and a cfg of 4.0, it raises the question of whether the performance gains shown in Fig 3 and Table 4 are meaningful, as it seems the model is unlikely to be near convergence at 500K samples. Indeed, the loss curves seem to suggest that even at 1.2M steps the model is far from convergence. Given the trend in performance visible in Fig 3, it looks possible that at 2M and cfg=4.0 the improvement of ProMoE may no longer substantive, but the authors haven't included these important results.

**Single dataset for experimental validation**. The authors present their experimental results on just ImageNet-1K. Though the authors do a good job of validating at multiple model sizes, the empirical contribution would be much more persuasive if the findings could be validated across multiple datasets.


[1] On the representation collapse of spare moe [Chi et al, NeurIPS 2022]
[2] Statistical advantages of perturbing cosine router in moe [Nguyen et al, ICLR 2025]
[3] Sparse moe are domain generalizable learners [Li et al, ICLR 2023]

**Questions:**

I'd strongly recommend the authors to include analysis of the kind seen in Table 4 but at training step=2M and cfg=4.0 across the MoE baselines and the dense baseline. This would provide a comprehensive analysis of the empirical benefits of ProMoE at convergence. Just choosing one size, ideally the largest, would be sufficient. If the authors can demonstrate the strong empirical results hold up at higher training step and cfg settings, I would consider raising my score, but for now it seems possible that the reported gains are too far from convergence to be meaningful.

If the authors could include an additional dataset that would also help boost the empirical contribution.

---

> ### Author Response · Authors · 2025-11-23
> **Rebuttal by Authors (1/3)**
>
> We sincerely thank you for recognizing our "_strong experimental results with uniform improvements_", "_comprehensive scalability analysis covering four model sizes_", and "_clear presentation_". Below, we address each of your questions and hope to clarify any concerns.
>
> ---
> **[W1] On the clarification of the technical novelty**
>
> We thank the reviewer for this comment. We have carefully read the mentioned works [1, 2, 3] and gained a deeper understanding of MoE designs in the NLP and discriminative vision tasks. However, **ProMoE is specifically designed to address the unique challenges of visual generation tasks**, leading to fundamental differences from these prior works. To fully address your concerns, we begin by detailing our technical novelties:
>
> 1. Unlike NLP or discriminative vision tasks in [1, 2, 3], DiT-based visual generation employs different model architectures and poses unique challenges, notably the **functional heterogeneity of visual tokens**. To address this, we design a novel **two-step router**. First, **Conditional Routing** explicitly separates tokens based on their functional roles (conditional vs. unconditional) in the diffusion process, which is important for the performance of visual MoE, as shown in ``Table 8 of the revised manuscript``.
> This functional separation enables the subsequent Prototypical Routing to perform more effective fine-grained expert assignment. This specific two-step routing design is absent in prior MoE methods.
>
> 2. Visual features for generation are **continuous and highly redundant**, making expert specialization much more challenging. Prior methods optimize routing only **implicitly** via the final task loss, which is often insufficient for robust expert specialization under high visual redundancy. To address this, we propose a novel **Routing Contrastive Loss (RCL)** that provides **explicit semantic routing guidance**. By actively pulling semantically similar tokens toward their assigned prototypes and pushing dissimilar ones away, RCL transforms routing from an implicit association into explicit metric learning, ensuring that experts focus on meaningful semantic patterns rather than low-level redundancy.
>
> Besides the core technical innovations discussed above, we further clarify specific differences in design choices compared to [1, 2, 3]:
>
> 1. The expert embeddings in [1, 2, 3] are standard learnable parameters **without explicit semantic constraints**. In contrast, our learnable prototypes are specifically designed and explicitly supervised by RCL to function as **semantic cluster centers** in the latent space, representing groups of semantically similar tokens and capturing **high-level semantic information**. This distinct property, absent in implicitly optimized embeddings of prior work, is crucial to enabling effective routing in the continuous visual space.
>
> 2. As the reviewer noted, our use of an **identity activation** differs from the standard non-linear functions in [1, 2, 3]. Our ablation studies (``Table 9 of the revised manuscript``) confirm that **preserving the raw similarity structure in latent space** via identity activation consistently yields superior performance.
> This choice is not isolated but synergizes with our two-step routing and RCL to form an effective framework specifically optimized for DiT-based generation.
> We believe these findings offer valuable insights for visual MoE design.
>
> Thank you again for your insightful comments, which also help us better distinguish our DiT-based MoE design from prior NLP MoE work. We have summarized our core contributions and innovations in the Global Comment: ``Summary of Core Contributions and New Experiments``. We respectfully hope the reviewer will consider these points holistically in the assessment of our work's technical novelty.

---

> ### Author Response · Authors · 2025-11-23
> **Rebuttal by Authors (2/3)**
>
> **[W2 & Q1] On the performance comparison of extended training steps and higher CFG settings.**
>
> We thank the reviewer for this insightful comment. We address your concerns from four aspects, supported by new experimental results:
>
> 1. **Substantial gains persist at 2M training steps, higher CFG (4.0), and different model sizes**.
> Following your suggestion, we evaluate L and XL ProMoE models against dense and SOTA DiffMoE baselines at 2M steps with CFG scales of 1.0, 1.5, and 4.0.
> As shown in the table below, our ProMoE models consistently outperform both baselines across all CFG settings and for both L/XL sizes, on both FID and IS.
>     - At CFG=4.0, the gains are especially clear: for L-size, ProMoE reduces FID by ~21% over the dense model and 10% over DiffMoE; for **XL-size, ProMoE reduces FID by ~30% over the dense model and 16% over DiffMoE**, while also achieving the best IS.
>     - At lower CFG values, ProMoE still yields consistent improvements over both dense and DiffMoE baselines: at CFG=1.5, it reduces FID by ~21% (L) and ~18% (XL) over the dense model, and by ~14% (L) and ~17% (XL) over DiffMoE; at CFG=1.0, it reduces FID by ~8% (L) and ~12% (XL) over the dense model.
>     - **ProMoE-L at 2M steps surpasses Dense/DiffMoE-XL on FID with fewer activated parameters**: even with 458M activated parameters (vs. 675M for XL), ProMoE-L already achieves better FID than both Dense-XL and DiffMoE-XL at CFG=4.0 and 1.5.
>
>     These results further solidify ProMoE's better performance and parameter efficiency under prolonged training and stronger guidance.
>
> **Table D1: Quantitative comparison under Rectified Flow on 2M training steps** on ImageNet (256×256).
>
> |Model|Activated Params.|Total Params.|cfg=1.0 FID50K ↓|cfg=1.0 IS ↑|cfg=1.5 FID50K ↓|cfg=1.5 IS ↑|cfg=4.0 FID50K ↓|cfg=4.0 IS ↑|
> |:-|:-:|:-:|:-:|:-:|:-:|:-:|:-:|:-:|
> |Dense-DiT-L-Flow (2M)|458M|458M|10.55|112.55|2.81|266.24|17.71|464.25|
> |DiffMoE-L-Flow (2M)|458M|1.163B|10.09|122.01|2.57|276.75|15.70|465.60|
> |**ProMoE-L-Flow (2M)**|458M|1.063B|**9.67**|**125.88**|**2.22**|**290.61**|**14.05**|**466.96**|
> |Dense-DiT-XL-Flow (2M)|675M|675M|9.30|121.92|2.55|282.59|17.05|474.39|
> |DiffMoE-XL-Flow (2M)|675M|1.735B|8.83|134.37|2.49|297.25|14.25|475.13|
> |**ProMoE-XL-Flow (2M)**|675M|1.568B|**8.16**|**139.66**|**2.08**|**304.26**|**11.95**|**478.60**|
>
> 2. **Results at 400K–500K steps are also meaningful and standard**.
> In fact, given limited computational resources, evaluating models at 400K–500K steps is a reasonable and widely used setting. Reporting such intermediate checkpoints is standard in the DiT literature (_e.g._, DiT[1], SiT[2], DiG[3]) and serves as a valid proxy for comparing training efficiency and architectural potential. In our case, the performance ranking at 500K steps is consistent with that at 2M steps, further supporting the reliability of these intermediate results.
>
> 3. **Superior training efficiency is also a key advantage of ProMoE**.
> ProMoE reaches a given performance level with substantially fewer training steps.
> For example, with CFG=1.5, ProMoE-L-Flow at 500K steps (FID 2.79) already outperforms Dense-DiT-L-Flow at 2M steps (FID 2.81). Similarly, the training loss curves in ``Figure 5`` show that reaching a loss of ~0.73 takes the dense model about 1.2M steps, but only ~400K steps for ProMoE, corresponding to roughly a **3× faster convergence**.
> This improved training efficiency is particularly valuable when scaling diffusion models.
>
> 4. **The improvement at 500K steps with CFG=1.5 is still substantial**.
> The gap at CFG=1.5 in ``Figure 3`` appears small due to **y-axis compression** when plotting widely different FID ranges. However, ``Table 5`` clearly shows that at 500K steps and CFG=1.5, Dense-DiT-L-Flow has FID 3.56, DiffMoE-L-Flow slightly improves this to 3.51 (~1.4% better), whereas ProMoE-L-Flow achieves 2.79, **a ~21.6% relative improvement over the dense model**. Thus, under the standard CFG=1.5 setting, ProMoE's gain is both substantial and clearly larger than that of DiffMoE.
>
> We sincerely thank you for your valuable comments, which further supporting the empirical validation of our approach near convergence.
>
> ---
> [1] Scalable Diffusion Models with Transformers
>
> [2] SiT: Exploring Flow and Diffusion-based Generative Models with Scalable Interpolant Transformers
>
> [3] DiG: Scalable and Efficient Diffusion Models with Gated Linear Attention

---

> > ### Author Response · Authors · 2025-11-27
> > **Rebuttal by Authors (3/3)**
> >
> > **[W3 & Q2] On the generalization of ProMoE to other datasets and modalities**
> >
> > We thank the reviewer for the invaluable feedback. Following your suggestion, we verify the generalization of ProMoE on text-to-image generation tasks, and evaluate it on the GenEval benchmark.
> >
> > For a comprehensive response, including the detailed experimental setup, quantitative comparisons on the GenEval benchmark, and qualitative results, **we kindly refer the reviewer to the Global Comment: ``Generalization Experiments on Text-to-Image Generation``**. The results presented there demonstrate that **ProMoE significantly outperforms both dense and competitive MoE baselines in the text-to-image task**, indicating robust generalization capabilities.
> >
> > ---
> > Finally, we sincerely thank you again for your time and constructive suggestions, which have strengthened the presentation and empirical results of our work.
> > We have incorporated the requested experiments into our revised manuscript in ``Section 5.2``, ``Tables 6 and 7``, and ``Figure 14``, with all new content highlighted in blue. We hope our responses can address your concerns, and we would be glad to address any further questions.

---

### Official Review · Reviewer_NvNP · 2025-10-30

**Soundness:** 3
**Presentation:** 3
**Contribution:** 3
**Rating:** 6
**Confidence:** 4

**Summary:**

This paper introduces ProMoE, a framework that successfully applies Mixture-of-Experts (MoE) to Diffusion Transformers (DiTs), addressing why previous attempts failed. The authors argue that visual tokens, unlike language tokens, have high redundancy and functional differences, which hinders expert specialization. ProMoE solves this with a novel two-step router that first separates tokens by function and then assigns them to experts based on semantic content using learnable prototypes. This guided approach, enhanced by a new contrastive loss, enables strong expert specialization and achieves state-of-the-art results on ImageNet.

**Strengths:**

- This paper clearly diagnoses the problem of vision MoE and proposes an innovative ProMoE to solve it.

- The ProMoE achieves validated, state-of-the-art results on the ImageNet benchmark.

- The presentation is clear and easy to understand.

**Weaknesses:**

- What is the fundamental difference between prototypical routing and conventional MoE routing mechanisms, such as one using a standard linear layer? The paper introduces "learnable prototypes", but this seems functionally very similar to using the learnable weights of a linear layer to calculate token-expert affinities. Could you clarify what makes this prototypical approach a genuine innovation, rather than just a conceptual re-framing of a standard linear gating mechanism?

- The routing mechanism in ProMoE appears to rely on pre-defined structures tailored for specific categories, unlike the autonomous expert specialization seen in LLMs. This raises questions about its generalizability—how would ProMoE handle open-ended conditional inputs, such as a natural-language prompt, rather than predefined categories? This design appears less flexible and general.

- How is the number of experts determined, and what is the rationale for that specific choice? Are there more detailed ablation studies on the impact of varying the number of experts?

**Questions:**

See Weaknesses.

---

> ### Author Response · Authors · 2025-11-23
> **Rebuttal by Authors (1/2)**
>
> We sincerely thank you for recognizing that our work "_clearly diagnoses the problem of vision MoE_", is "_innovative_", "_achieves state-of-the-art results_" and offers a "_clear presentation_". Below, we address each of your questions and hope to clarify any concerns.
>
> ---
> **[W1] On the clarification of fundamental differences between prototypical routing and conventional linear gating**
>
> We thank the reviewer for this insightful question. The term "prototype" is not a cosmetic rephrasing; it reflects the specific role the routing weights play in our prototypical routing.
> We would like to clarify that **our prototypical routing is not just about introducing learnable prototypes**. It also **includes a dedicated assignment mechanism, activation mechanism, and optimization objective**, which fundamentally differ from standard linear gating and are specifically designed to address the high redundancy of visual tokens in DiTs. We elaborate on these aspects below.
>
> 1. **Assignment Mechanism**. Unlike standard linear gating which relies on dot products, we strictly normalize both input tokens and learnable prototypes, utilizing cosine similarity for expert assignment.
> **This design grounds the routing process within a semantic latent space, enabling each prototype to represent a distinct semantic class or feature cluster**, unlike the unstructured weights used in standard linear gating.
>
> 2. **Activation Mechanism**. Conventional linear gating typically uses Softmax activation, which enlarges score differences, causing a few experts to receive high routing weights while others are almost ignored. In contrast, we use **Identity activation to preserve the original similarities in the semantic space** without such amplification. This leads to more stable training and superior performance compared to standard gating, as shown in ``Table 9 of the revised manuscript``.
>
> 3. **Optimization Objective**. Standard linear gating is optimized **implicitly**, relying solely on the final diffusion loss. In contrast, our **prototypical routing supports explicit semantic routing guidance** (_e.g._, our proposed **Routing Contrastive Loss**). Because prototypes serve as semantic cluster centers, Routing Contrastive Loss can directly optimize the routing process by pulling semantically similar tokens closer to their prototypes and pushing dissimilar tokens and prototypes apart. This explicit optimization significantly strengthens expert specialization and overall performance.
>
> In summary, these three components form a holistic routing method that is fundamentally distinct from standard linear gating. Our extensive experiments demonstrate its superior effectiveness, and we believe this approach offers valuable insights for the design of future visual MoEs.

---

> > ### Author Response · Authors · 2025-11-23
> > **Rebuttal by Authors (2/2)**
> >
> > **[W2] On the generalization of ProMoE to open-ended inputs**
> >
> > We sincerely thank the reviewer for the invaluable feedback. We first respectfully clarify a key **misunderstanding**: **ProMoE is category-agnostic and achieves specialization fully autonomously, similar to MoE in LLMs, without relying on any pre-defined categories**. This is a core advantage over classification-based or clustering-based routing methods (Sec. 4.2), which typically require category priors.
> >
> > Specifically, our conditional routing separates tokens solely based on whether conditioning is present or absent, independent of image categories. For prototypical routing, prototypes are initialized without any category priors and are learned end-to-end via the diffusion and routing contrastive losses, without category-level supervision. Therefore, **our routing design is flexible and naturally supports open-ended inputs such as textual prompts**, since the routing mechanism remains unchanged and is driven by adaptive semantic similarity rather than predefined structures.
> >
> > To further validate this generalizability, we conduct new text-to-image experiments, evaluated on the GenEval benchmark. For a comprehensive response, including the detailed experimental setup, quantitative comparisons on the GenEval benchmark, and qualitative results, **we kindly refer the reviewer to the Global Comment: ``Generalization Experiments on Text-to-Image Generation``**. The results presented there demonstrate that **ProMoE significantly outperforms both dense and competitive MoE baselines in the text-to-image task**, indicating robust generalization capabilities.
> >
> > We have also incorporated these new results into the revised manuscript, specifically in ``Section 5.2``, ``Table 6``, and ``Figure 14``, with all added content highlighted in blue.
> >
> > ---
> > **[W3] On the clarification of the chosen expert number and the impact of varying the number of experts**
> >
> > We thank the reviewer for this constructive question. **The choice of the expert number reflects a trade-off between performance and total parameter count**: more experts typically improve performance but also increase model size and training cost. In our main experiments, we set the number of experts to 14 to match the total parameter count of the MoE baselines, ensuring a fair comparison under comparable model sizes.
> >
> > In fact, **we have conducted ablation studies on the impact of varying the number of experts, as shown in ``Figure 3(c) and Figure 10 of the revised manuscript``**. ProMoE exhibits excellent scalability, with performance consistently improving as the number of experts increases. For clarity, we summarize these results in the table below.
> >
> > **Table C1: Ablation of varying expert number on ProMoE-L-Flow**, trained on ImageNet (256×256) for 500K steps, evaluated with CFG=1.0.
> >
> > |Number of Experts|6|10|14|18|
> > |:-|:-:|:-:|:-:|:-:|
> > |FID 50K↓|12.42|12.03|11.61|11.35|
> > |IS ↑|97.40|98.96|100.82|102.63|
> >
> > Notably, **ProMoE outperforms state-of-the-art MoE baselines while using fewer experts and lower total parameter counts**. For instance, with only 6 experts (0.66B total parameters), ProMoE achieves an FID of 12.42 (CFG=1.0), surpassing DiffMoE which uses 8 experts and nearly double the parameters (1.095B parameters, FID 14.46). This highlights the superior parameter efficiency and effectiveness of our approach.
> >
> > We also conduct a new experiment varying the expert number for K-means-based routing (unconditional routing is not used here). As shown in the table below, K-means-based routing is highly sensitive to the expert number and fails to scale effectively: increasing experts yields fluctuating performance with no clear gain, despite the increased parameter count. This contrast further highlights the scalability and robustness of our ProMoE compared to clustering-based routing.
> >
> > **Table C2: Ablation of varying expert number in K-Means-based Routing on base model size**, trained with Rectified Flow on ImageNet (256×256) for 500K steps, evaluated with CFG=1.5. The setup uses 1 activated expert and 1 shared expert.
> >
> > |Number of Experts|3|5|9|13|21|
> > |:-|:-:|:-:|:-:|:-:|:-:|
> > |FID 10K↓|12.55|11.52|12.43|11.41|12.15|
> >
> > ---
> > Finally, we sincerely thank you again for your time and constructive suggestions, which have strengthened the clarity, robustness, and empirical validation of our work. We have incorporated the requested experiments into our revised manuscript in ``Section 5.2 and Appendix Section D.3``, ``Tables 6 and 16``, and ``Figure 14``, with all new content highlighted in blue. We hope our responses can address your concerns, and we would be glad to address any further questions.

---

### Official Review · Reviewer_1uA8 · 2025-10-31

**Soundness:** 3
**Presentation:** 4
**Contribution:** 3
**Rating:** 6
**Confidence:** 4

**Summary:**

The paper proposes ProMoE, a novel MoE framework for Diffusion Transformers that addresses the failure of prior MoE designs in vision via a two-step router with explicit routing guidance. It introduces conditional routing to separate functional roles and prototypical routing with learnable prototypes, enhanced by a routing contrastive loss.

**Strengths:**

1. The paper effectively addresses the core challenges of visual token redundancy and functional heterogeneity in Diffusion Transformers, introducing mechanisms that enable true expert specialization within the Mixture-of-Experts framework.
2. The proposed method demonstrates strong and consistent scaling behavior across multiple model sizes, validating its robustness and efficiency under both Rectified Flow and DDPM training paradigms.

**Weaknesses:**

1. The experiments are conducted solely on ImageNet-1K for class-conditional generation, without evaluations on other datasets or modalities, which limits the evidence of generalization.
2. The paper does not report quantitative expert utilization, such as the proportion of tokens or capacity per expert, making it hard to assess balance and specialization.

**Questions:**

1.Could the authors provide quantitative statistics of expert utilization (e.g., token-per-expert ratios or activation entropy) to substantiate the claimed specialization and balance?
2.Could the authors compare ProMoE with other unsupervised clustering methods that support top-K routing, such as GMM or deep clustering?

---

> ### Author Response · Authors · 2025-11-23
> **Rebuttal by Authors (1/2)**
>
> We sincerely thank you for recognizing that our approach "_effectively addresses the core challenges of visual token redundancy and functional heterogeneity in DiTs_", "_enables true expert specialization within the MoE framework_", "_demonstrates strong and consistent scaling behavior_", and "_robust and efficient performance under two paradigms_". Below, we address each of your questions and hope to clarify any concerns.
>
> ---
> **[W1] On the generalization of ProMoE to other datasets and modalities**
>
> We thank the reviewer for this invaluable feedback. Following your suggestion, we have conducted new text-to-image experiments to further demonstrate the generalization ability of ProMoE.
>
> For a comprehensive response, including the detailed experimental setup, quantitative comparisons on the GenEval benchmark, and qualitative results, **we kindly refer the reviewer to the Global Comment: ``Generalization Experiments on Text-to-Image Generation``**. The results presented there demonstrate that **ProMoE significantly outperforms both dense and competitive MoE baselines in the text-to-image task**, indicating robust generalization capabilities.
>
> We have also incorporated these new results into the revised manuscript, specifically in ``Section 5.2``, ``Table 6``, and ``Figure 14``, with all added content highlighted in blue.
>
> ---
> **[W2 & Q1] On the quantitative analysis of expert utilization**
>
> We thank the reviewer for this constructive suggestion. Following your advice, we report the quantitative expert utilization (token-per-expert ratios) for ProMoE compared to the DiT-MoE baseline.
>
> We report token-per-expert ratios on two disjoint subsets, each consisting of 200 randomly sampled classes. For each subset, we randomly generate 10,000 images and compute the proportion of tokens assigned to each expert. We then average these proportions over all layers for each expert.
>
> As shown in ``Figure 6 of the revised manuscript`` and the table below, DiT-MoE exhibits very similar token-per-expert distributions across different class subsets. Within each subset, the number of tokens assigned to different experts is also very close, with no clear differentiation. This suggests that the experts are poorly specialized and that the routing fails to induce meaningful expert diversity.
>
> In contrast, **our method exhibits clear expert specialization**:
> 1) It produces distinct expert utilization patterns across disjoint class subsets, confirming effective expert specialization for different inputs.
> 2) Within each subset, some experts are mildly favored (_i.e._, selected more frequently than average), while others are used less often, yet no expert is starved or overused.
> These observations suggest that ProMoE achieves meaningful expert specialization while maintaining good load balancing.
>
> For quantitative reference, we provide the specific token-per-expert ratios in the table below. However, we strongly encourage reviewer to refer to ``Figure 6 of the revised manuscript`` for a more intuitive visualization of these distinct distribution patterns.
>
> **Table B1: Quantitative analysis of expert utilization**.
>
> |Method & Subset|E0|E1|E2|E3|E4|E5|E6|E7|E8|E9|E10|E11|
> |:-|:-:|:-:|:-:|:-:|:-:|:-:|:-:|:-:|:-:|:-:|:-:|:-:|
> |**DiT-MoE-L-Flow** (Class Subset 1)|12.6%|11.4%|13.0%|11.4%|12.8%|13.5%|11.9%|13.4%|-|-|-|-|
> |**DiT-MoE-L-Flow** (Class Subset 2)|12.9%|11.1%|12.9%|11.5%|12.7%|13.5%|12.0%|13.2%|-|-|-|-|
> |**ProMoE-L-Flow** (Class Subset 1)|7.8%|8.3%|9.7%|9.8%|10.9%|7.4%|8.1%|4.8%|7.7%|6.1%|10.6%|8.8%|
> |**ProMoE-L-Flow** (Class Subset 2)|5.5%|8.6%|12.4%|9.0%|9.3%|7.6%|9.7%|5.6%|6.7%|5.6%|11.4%|8.6%|

---

> > ### Author Response · Authors · 2025-11-23
> > **Rebuttal by Authors (2/2)**
> >
> > **[Q2] On the comparison of other unsupervised clustering methods.**
> >
> > Thank you for your insightful suggestion.
> > Following your advice, we conduct new experiments comparing our ProMoE with a GMM-based routing baseline on the large (L) model. The results in the table below show that
> > **ProMoE consistently outperforms the GMM-based routing baseline** across all metrics (FID and IS) and CFG scales under both top-1 (E14A1) and top-3 (E14A3) activation settings.
> > We argue that GMM, like other clustering-based routing schemes, relies purely on implicit learning and lacks explicit guidance, which is insufficient for robust expert specialization in vision MoE, especially for large-scale models. In contrast, ProMoE achieves better performance, highlighting the robustness and effectiveness of our explicit semantic routing guidance strategies.
> >
> > Furthermore, we would like to clarify that both the K-means-based and GMM-based routing baselines can be regarded as specific instances of deep clustering. They use the DiT block as a feature extractor and iteratively perform unsupervised clustering on deep features to guide routing. If the reviewer has a particular deep clustering algorithm in mind, we would be happy to include further comparisons.
> >
> > **Table B2: Quantitative comparison with GMM-based routing baseline under Rectified Flow** on ImageNet (256×256) after 500K training steps, evaluated with CFG scales of 1.0 and 1.5.
> >
> > |Model (500K)|Experts|Activated Params.|Total Params.|cfg=1.0 FID50K ↓|cfg=1.0 IS ↑|cfg=1.5 FID50K ↓|cfg=1.5 IS ↑|
> > |:-|:-:|:-:|:-:|:-:|:-:|:-:|:-:|
> > |GMM-based Routing|E14A1S1U1|458M|1.063B|15.56|84.94|3.76|206.05|
> > |**ProMoE-L-Flow**|E14A1S1U1|458M|1.063B|**11.61**|**100.82**|**2.79**|**244.21**|
> > |GMM-based Routing|E14A3S1U1|558M|1.063B|15.44|86.09|3.72|208.15|
> > |**ProMoE-L-Flow**|E14A3S1U1|558M|1.063B|**11.40**|**103.78**|**2.72**|**246.96**|
> >
> > ---
> > Finally, we would like to sincerely thank you again for your time and valuable comments regarding experimental completeness. The additional experiments you suggested have further solidified the evidence for the effectiveness, generalization capability, and expert specialization of our method. We have incorporated these requested experiments and analyses into our revised manuscript in ``Section 5.2 and Appendix Section C.8``, ``Tables 6 and 15``, and ``Figures 6 and 14``, with all new content highlighted in blue. We hope our responses can address your concerns, and we would be glad to address any further questions.

---

### Official Review · Reviewer_4DXQ · 2025-11-01

**Soundness:** 3
**Presentation:** 3
**Contribution:** 3
**Rating:** 6
**Confidence:** 2

**Summary:**

This paper introduce a new expert routing method for Diffusion Transformers, by treating conditional and unconditional tokens independently. Routing guidance and contrastive learning are further introduced to enhance the performance.

**Strengths:**

1. The rationale for separating conditional and unconditional tokens is clear and well-founded.
2. The investigation into routing guidance and load balancing is insightful and valuable.

**Weaknesses:**

1. It would be beneficial to include ablation studies on dense models with conditional routing to determine whether the performance gain stems solely from conditional routing itself or requires combination with routing enhancements.

2. Since one key advantage of MoE models is improved computational efficiency, the authors are encouraged to report training and inference times, as well as FLOPs, in comparison to both dense models and other MoE variants.

**Questions:**

NA

---

> ### Author Response · Authors · 2025-11-23
> **Rebuttal by Authors**
>
> We sincerely thank you for recognizing our "_clear, well-founded rationale for conditional routing_" and "_insightful and valuable analysis of routing guidance_." Below, we address each of your questions and hope to clarify any concerns.
>
> ---
> **[W1] On the ablation study of conditional routing on dense models**
>
> We thank the reviewer for this valuable feedback. Following your suggestion, we conduct an ablation study on a dense model augmented only with conditional routing.
> The results in the following table demonstrate that while **conditional routing alone provides a performance improvement, the gains are more significant when combined with our routing enhancements** (prototypical routing and routing contrastive loss).
> This is because prototypical routing augments conditional routing with a more flexible and effective expert-selection mechanism, while routing contrastive loss further explicitly promotes intra-expert coherence and inter-expert diversity.
> This finding underscores that the full performance benefit stems from the synergistic effect of our complete design, validating the critical contribution of each proposed component.
>
> **Table A1: Ablation study of conditional routing** on ImageNet (256×256) after 500K training steps, trained with Rectified Flow and evaluated with CFG scales of 1.0 and 1.5.
>
> |Model (500K)|cfg=1.0 FID50K ↓|cfg=1.0 IS ↑|cfg=1.5 FID50K ↓|cfg=1.5 IS ↑|
> |:-|:-:|:-:|:-:|:-:|
> |Dense-DiT-B-Flow|30.61|49.89|9.02|131.13|
> |+ Conditional Routing|28.36|53.01|8.50|135.05|
> |+ Prototypical Routing and RCL|**24.44**|**60.38**|**6.39**|**154.21**|
>
> ---
> **[W2] On the comparison of computational cost and efficiency**
>
> We thank the reviewer for this insightful suggestion. Following your advice, we report the computational cost of our model in terms of training time, inference time, and FLOPs, and compare it against dense and MoE baselines.
> The inference time measures the core denoising process (excluding VAE decoding) with a batch size of 128, a classifier-free guidance (CFG) scale of 1.5, and 250 sampling steps. For a fair comparison, all experiments are conducted on 4 NVIDIA A800 GPUs under identical hardware settings.
>
> The results in the table below show that **ProMoE achieves lower inference time and fewer GFLOPs than the SOTA MoE method DiffMoE**, and maintains GFLOPs comparable to other MoE baselines while **delivering substantially higher performance**, as shown in ``Table 5 of the revised manuscript``. This demonstrates that the performance gains primarily stem from our superior methodological design.
>
> **Table A2: Comparison of Training Time, Inference Time, and FLOPs with Dense Model and MoE baselines under Rectified Flow** on ImageNet (256×256) after 500K training steps.
>
> |Model (500K)|#Experts|# Activated Params.|# Total Params.|# Training Time|# Inference Time (cfg=1.5)|# GFLOPs|
> |:-|:-:|:-:|:-:|:-:|:-:|:-:|
> |Dense-DiT-L-Flow|-|458M|458M|166.67 GPU Hours|1.25 sec/sample|77.50|
> |DiT-MoE-L-Flow|E8A1S0U0|458M|1.163B|333.33 GPU Hours|1.49 sec/sample|77.50|
> |EC-DiT-L-Flow|E8A1S0U0|458M|1.163B|277.78 GPU Hours|1.49 sec/sample|77.50|
> |DiffMoE-L-Flow|E8A1S0U0|458M|1.095B|333.33 GPU Hours|1.64 sec/sample|82.53|
> |**ProMoE-L-Flow**|E14A1S1U1|458M|1.063B|333.33 GPU Hours|1.53 sec/sample|77.72|
>
> ---
> Finally, we would like to sincerely thank you again for your time and valuable comments on the experimental completeness. The additional experiments you suggested further validate the effectiveness of our modules and strengthen the advantages of our method. We have incorporated the requested experiments and analyses into our revised manuscript in ``Sections 5.2 and 5.4`` and ``Tables 5, 8 and 14``, with all new content highlighted in blue. We hope our responses can address your concerns, and we would be glad to address any further questions.

---

### Author Response · Authors · 2025-11-23
**Generalization Experiments on Text-to-Image Generation**

We thank Reviewers 1uA8, NvNP, and vTUm for their valuable constructive feedback regarding the generalization ability of ProMoE beyond Class-to-Image generation on ImageNet. We fully agree that demonstrating robust generalization to other datasets and modalities is crucial.

In response to their suggestions, we have conducted new experiments on the challenging Text-to-Image (T2I) generation task. Below, we first summarize the experimental setup and then present the results. Full details of experimental setup are provided in ``Appendix A of the revised manuscript``.

We adopt an MM-DiT–style architecture for effective conditioning. The dense baseline uses 36 layers, 16 attention heads, a hidden size of 11,008, and has 3B parameters. On top of this backbone, we consider two MoE variants: _1)_ a standard LLM-style MoE baseline, Token-Choice MoE, with 5 standard experts and 12B total parameters (3B activated), and _2)_ our ProMoE, which has the same total and activated parameter counts with 5 experts (1 unconditional and 4 standard), ensuring a fair comparison.
All models are trained on a 2M-image subset of LAION-5B to enable rapid validation, using identical training and inference configurations to ensure a fair comparison. We use the pretrained Wan2.5-Preview VAE for image encoding and Qwen2.5-VL as the text encoder.

We evaluate our method on the GenEval benchmark, as shown in the table below. **ProMoE significantly outperforms both the dense baseline and the Token-Choice MoE across the overall metric and all sub-tasks**, with a remarkable relative improvement of 18.7% over the dense model.
Specifically, ProMoE improves the Overall score from the dense baseline's 0.390 to 0.463. While the Token-Choice MoE shows a modest overall improvement, it suffers from performance degradation on some sub-tasks like "Counting". In contrast, ProMoE outperforms the Token-Choice MoE on every single metric. This not only demonstrates the superior performance of our ProMoE but also highlights its robust generalization capabilities across more challenging generation tasks.

We also provide qualitative results in ``Figure 14 of the updated manuscript``, further verifying ProMoE's capability in high-quality text-to-image generation.

**Table G1: Quantitative comparison on text-to-image benchmark (GenEval) under Rectified Flow** after 400K training steps.

|Model (400K)|Experts|Activated Params.|Total Params.|Single Obj.↑|Two Obj.↑|Counting↑|Colors↑|Position↑|Color Attr.↑|**Overall**↑|
|:-|:-:|:-:|:-:|:-:|:-:|:-:|:-:|:-:|:-:|:-:|
|Dense Model|-|3B|3B|0.840|0.275|0.362|0.611|0.095|0.155|0.390|
|Token-Choice MoE|E5A1S0U0|3B|12B|0.856|0.320|0.334|0.627|0.157|0.207|0.417|
|**ProMoE**|E5A1S0U1|3B|12B|**0.884**|**0.371**|**0.418**|**0.675**|**0.212**|**0.217**|**0.463**|

We once again express our sincere gratitude for the reviewers' valuable comments. The additional generalization experiments further validate the superiority and robustness of our ProMoE. We have incorporated these comprehensive analyses into the revised manuscript in ``Section 5.2``, ``Table 6``, and ``Figure 14``, with all new content highlighted in blue. We hope our responses adequately address the concerns about generalization, and we would be glad to answer any further questions.

---

### Author Response · Authors · 2025-11-23
**Summary of Core Contributions and New Experiments**

We sincerely thank the area chairs and all reviewers for their time, constructive feedback, and recognition of our work. We are encouraged that the reviewers recognized our "_clear problem diagnosis and innovative solution_" (Reviewers 1uA8, NvNP), "_strong, state-of-the-art experimental results_" (Reviewers 1uA8, NvNP, vTUm), "_comprehensive scalability analysis across model sizes_" (Reviewers 1uA8, vTUm), "_clear presentation_" (Reviewers NvNP, vTUm), "_the well-founded rationale for conditional routing_" (Reviewer 4DXQ), "_insightful and valuable investigation of routing guidance_" (Reviewer 4DXQ), "_method enables true expert specialization_" (Reviewer 1uA8). Below, we summarize our core contributions and technical novelties:

1. We **visualize and analyze the fundamental differences between visual and language tokens** (e.g., ``Figures 1, 11, and 12 of the revised manuscript``), identifying that the suboptimal performance of standard MoE in DiTs arises from the **functional heterogeneity** and **high spatial redundancy** unique to visual tokens. This insight establishes the necessity of explicit routing guidance for DiT-based MoE architectures.

2. We propose a novel **two-step router** specifically designed for visual generation. It first tackles functional heterogeneity by separating tokens via **conditional routing**, and then assigns experts based on semantic content via **prototypical routing**. We further **systematically study two semantic guidance strategies**, demonstrating that both explicit (classification-based) and implicit (clustering-based) guidance lead to clear improvements.

3. We propose a novel **Routing Contrastive Loss (RCL)** based on learnable prototypes. Unlike standard MoE routers that rely on implicit optimization via the final task loss, RCL transforms routing from implicit association into **explicit metric learning**, enforcing robust expert specialization to effectively handle highly redundant visual tokens.

4. Our extensive experiments demonstrate that ProMoE **consistently outperforms** dense baselines and SOTA MoE methods across both Rectified Flow and DDPM paradigms. We further validate the method's robustness through comprehensive scalability tests, strong generalization on text-to-image generation tasks, and extended training analysis up to 2M steps.

Additionally, we have included extensive new experiments in the revised manuscript, with key additions highlighted in blue:

1. New ablation study on the impact of conditional routing on dense models in ``Section 5.4`` and ``Table 8`` (suggested by Reviewer 4DXQ).

2. New quantitative analysis of training time, inference time, and FLOPs in ``Section 5.2`` and ``Tables 5 and 14`` (suggested by Reviewer 4DXQ).

3. New experimental results on text-to-image generation tasks, including both quantitative comparisons and qualitative visualizations, in ``Section 5.2``, ``Table 6``, and ``Figure 14`` (suggested by Reviewers 1uA8, NvNP, vTUm).

4. New quantitative analysis of expert utilization in ``Section 5.2`` and ``Figure 6`` (suggested by Reviewer 1uA8).

5. New comparison results with additional unsupervised clustering methods in ``Appendix Section C.8`` and ``Table 15`` (suggested by Reviewer 1uA8).

6. New ablation study regarding the number of experts in K-Means-based Routing in ``Appendix Section D.3`` and ``Table 16`` (suggested by Reviewer NvNP).

7. New comparison results of performance over extended training steps and higher CFG in ``Section 5.2`` and ``Table 7`` (suggested by Reviewer vTUm).

We sincerely thank the area chairs and reviewers again for their invaluable time and effort.

---

### Meta-Review · Area_Chair_68Vv · 2026-01-07

**Summary:**

This paper proposes an expert routing method for Diffusion Transformers, by treating conditional and unconditional tokens independently. The four reviewers pointed out multiple concerns about different aspects of this paper, including but not limited to:

1. The technical novelty is limited. The learnable prototypes appear to just be standard learnable expert embeddings and semantic similarity is computed using a cosine similarity between inputs and expert embeddings. The authors use an identity activation function which is non-standard, but this alone does not really qualify the entire routing mechanism as novel.
2. Some experimental results are missing. It raises the question of whether the performance gains shown in Fig 3 and Table 4 are meaningful, as it seems the model is unlikely to be near convergence at 500K samples. Given the trend in performance visible in Fig 3, it looks possible that at 2M and cfg=4.0 the improvement of ProMoE may no longer substantive, but the authors haven't included these important results.
3. The authors present their experimental results on just ImageNet-1K. The empirical contribution would be much more persuasive if the findings could be validated across multiple datasets.
4. It would be beneficial to include ablation studies on dense models with conditional routing to determine whether the performance gain stems solely from conditional routing itself or requires combination with routing enhancements.
5. The authors are encouraged to report training and inference times, as well as FLOPs, in comparison to both dense models and other MoE variants.
6. The paper does not report quantitative expert utilization, such as the proportion of tokens or capacity per expert, making it hard to assess balance and specialization.
7. The fundamental difference between prototypical routing and conventional MoE routing mechanisms is unclear. The authors are suggested to clarify what makes this prototypical approach a genuine innovation, rather than just a conceptual re-framing of a standard linear gating mechanism.

Most of the concerns are probably addressed by the rebuttal. I recommend acceptance and encourage the authors to incorporate the rebuttal in the final version as promised.

**Reviewer Concerns:**

The 2nd to the7th concerns are partially addressed, while the 1st concern is not well addressed.

**Reviewer Scores:**

vTUm

---

### Decision · Program_Chairs · 2026-01-26

Accept (Poster)